

# Yukawa-Lorentz symmetry of interacting non-Hermitian birefringent Dirac fermions

**Sk Asrap Murshed and Bitan Roy**

Department of Physics, Lehigh University, Bethlehem, Pennsylvania, 18015, USA

## Abstract

The energy spectra of linearly dispersing gapless spin-3/2 Dirac fermions display birefringence, featuring two effective Fermi velocities, thus breaking the space-time Lorentz symmetry. Here, we consider a non-Hermitian (NH) generalization of this scenario by introducing a masslike anti-Hermitian birefringent Dirac operator to its Hermitian counterpart. At the microscopic level, a generalized $\pi$-flux square lattice model with imbalance in the hopping amplitudes in the opposite directions among spinless fermions between the nearest-neighbor sites gives rise to pseudospin-3/2 NH Dirac fermions in terms of internal, namely sublattice, degrees of freedom. The resulting NH operator shows real eigenvalue spectra over an extended NH parameter regime, and a combination of nonspatial and discrete rotational symmetries protects the gapless nature of such quasiparticles. However, at the brink of dynamic mass generation, triggered by Hubbardlike local interactions, the birefringent parameter always vanishes under coarse grain due to the Yukawa-type interactions with scalar bosonic order-parameter fluctuations. The resulting quantum critical state is, therefore, described by two decoupled copies of spin-1/2 Dirac fermions with a unique terminal Fermi velocity, which is equal to the bosonic order-parameter velocity, thereby fostering an emergent space-time Lorentz symmetry. Furthermore, depending on the internal algebra between the anti-Hermitian birefringent Dirac operator and the candidate mass order, the system achieves the emergent Yukawa-Lorentz symmetry either by maintaining its non-Hermiticity or by recovering a full Hermiticity. We discuss the resulting quantum critical phenomena and possible microscopic realizations of the proposed scenarios.

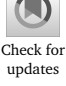
# 1 Introduction

The linear energy-momentum correspondence for gapless spin-1/2 Dirac fermions manifests a space-time Lorentz symmetry in conventional Dirac materials due to a unique Fermi velocity in such systems [1–5]. Spin-3/2 nodal Dirac fermions, on the other hand, display birefringent linear energy-momentum spectrum with two effective Fermi velocities [6–19]. Consequently, they spoil the space-time Lorentz symmetry. However, strong Hubbardlike local interactions, for example, bring birefringent Dirac fermions at the shore of dynamic mass generation across which the system becomes an insulator via a spontaneous symmetry breaking. Then the emergent quantum critical state enjoys a space-time Lorentz symmetry in terms of a unique terminal velocity of all the participating degrees of freedom in the following way. Due to the strong Yukawa-type interactions between gapless fermionic and bosonic order-parameter degrees of freedom, the birefringent parameter in the fermionic two-point correlation function always vanishes. At the same time, the unique Fermi velocity and the velocity of scalar bosonic fields achieve a common terminal value in the deep infrared regime. This outcome turns out to be insensitive to the actual nature of the candidate insulating ground state [14, 18].

While the emergent Lorentz symmetry in spin-3/2 Dirac systems has received a limited attention so far, a similar outcome holds in spin-1/2 Dirac systems quite generically [20–28], when the the interacting bosonic degrees of freedom represent (a) scalar order-parameter fields and (b) helicity-1 vector photon fields. More recently, the notion of the emergent Lorentz symmetry has been extended to spin-1/2 non-Hermitian (NH) Dirac systems interacting with bosonic fields, mediated by either order-parameter fluctuations or photons, when these systems in addition are coupled to a surrounding bath or the environment [29, 30]. The present discussion is aimed to establish the jurisdiction of emergent Lorentz symmetry in NH spin-3/2 birefringent Dirac systems, residing at the brink of spontaneous mass generation. Although, here we arrive at these conclusions by considering the continuum limit of a specific microscopic model, they should be endowed with far reaching consequences. For example, our work establishes the concept of Lorentz symmetry as an emergent phenomenon near NH quantum critical points, when the system resides at the verge of a mass ordering, which should be impervious to microscopic details and the nature of the ordered state. In addition, we show that such elegant spacetime symmetry can be achieved in quasi-relativistic systems even when they are coupled to environment or a bath, while maintaining such a coupling or by effectively decoupling from the bath. These results, thus, give us the luxury to conjecture emergent space-

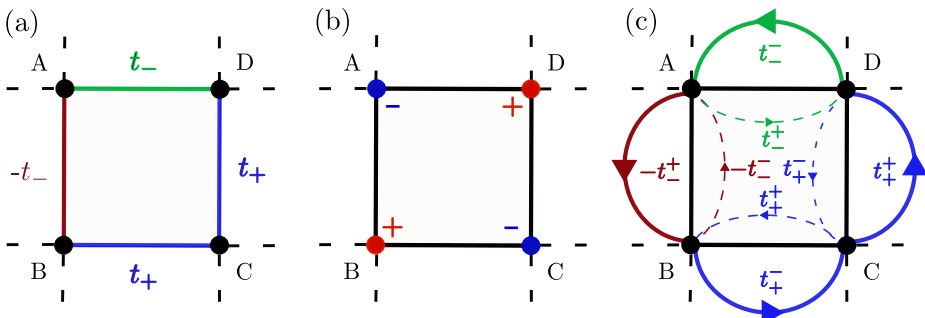

Figure 1: (a) A four-site unit-cell of a generalized $\pi$-flux square lattice, constituted by the sublattices A, B, C, and D, supporting gapless birefringent Dirac fermions as low energy excitations at half-filling around the $\mathbf{K} = (\pm\pi, \pm\pi)/(2a)$ points in the Brillouin zone, where $a$ is the lattice spacing and $t_\pm = t(1\pm\beta)$ with $t$ bearing the dimension of the hopping amplitude and $\beta$ representing the dimensionless birefringent parameter [7,8]. (b) A mass order for birefringent Dirac fermions, resulting from the quadrupolar arrangement of average fermionic density about the half-filling, where $+$ and $-$ represent its enhancement and depletion of equal magnitude away from the charge neutrality. (c) Hopping amplitudes between the nearest-neighbor sites in the directions of the arrows, giving rise to gapless non-Hermitian birefringent Dirac fermions, where $t_+^\pm = t_+(1\pm\alpha)$ and $t_-^\pm = t_-(1\pm\alpha)$. Here, the dimensionless parameter $\alpha$ captures the strength of the non-Hermiticity, leading to an imbalance in the hopping amplitudes in the opposite direction between each pair of the nearest-neighbor sites. The black dashed lines represent translations of the four-site unit-cell in the $x$ and $y$ directions.

time Lorentz symmetry in any system, where the energy-momentum relation is linear but the bands transform under arbitrary spin-1/2 representation, which in future can be established from complementary field theoretic and numerical studies.

## 1.1 A brief summary of key results

The effective low-energy model for birefringent spin-3/2 Dirac fermions is described by two copies of massless Dirac Hamiltonian that mutually commute with each other. The resulting Dirac operator thus features two effective Fermi velocities and its energy spectrum scales linearly with the momentum. Such a collection of effective spin-3/2 Dirac fermions in two spatial dimensions can be realized on a generalized $\pi$-flux square lattice with nearest-neighbor hopping processes among spinless fermions, for which the unit cell is constituted by four sites [7,8]. A NH birefringent Dirac operator is then constructed by supplementing its Hermitian counterpart by a *masslike* anti-Hermitian birefringent Dirac operator that also scales linearly with momentum. The anti-Hermitian operator is called masslike as it fully anticommutes with the Hermitian Hamiltonian, even though it vanishes linearly with momentum. Over an extended NH parameter regime, such an operator accommodates real eigenvalue spectrum. In the generalized $\pi$-flux square lattice model, the simplest realization of NH birefringent Dirac fermions results from a hopping imbalance in the opposite directions between each pair of nearest-neighbor sites. These constructions are shown in Fig. 1. The nodal or gapless nature of such emergent NH quasiparticles is protected by a combination of non-spatial and four-fold rotational symmetries, forbidding nucleation of any uniform mass order, otherwise leading to an insulation of the system, without the spontaneous breaking of any symmetry, as summarized in Table 1. Throughout, we consider spinless fermion, and the term 'spin' corresponds to 'pseudo-spin', resulting from the internal sublattice degrees of freedom of the system.

Here, we show that when a collection of NH spin-3/2 birefringent Dirac fermions live in the proximity to a dynamic mass generation via a spontaneous symmetry breaking, their Yukawa-type interaction with the scalar bosonic order-parameter fields generically gives birth to an emergent Lorentz symmetry in terms of a unique terminal velocity of all the participating degrees of freedom irrespective of the actual nature of the ordered state in the insulating side of the underlying quantum phase transition. We arrive at this conclusion from a leading-order (one-loop) renormalization group (RG) analysis of the associated Gross-Neveu-Yukawa quantum field theory, captured by the Feynman diagrams in Fig. 2, controlled by an $\varepsilon$ expansion about the upper critical three spatial dimensions where the theory becomes marginal with $\varepsilon = 3 - d$. However, depending on the internal Clifford algebra between the candidate mass order parameter and the anti-Hermitian component of the NH birefringent Dirac operator, the system accomplishes the Yukawa-Lorentz symmetry either by maintaining the non-Hermiticity (resulting from a coupling with an external bath) or by gaining full Hermiticity by decoupling itself from the environment. The results are summarized in terms of the RG flows of various velocity and birefringent parameters in Fig. 3. We name the former scenario a non-Hermitian Yukawa-Lorentz symmetry, while the latter one closely mimics the situation realized in closed or Hermitian systems. These outcomes, therefore, strongly suggest a generic emergent Lorentz symmetry in strongly correlated open Dirac materials, transforming under arbitrary spin-half representation, which we leave as a conjecture. Furthermore, we conclude that similar outcomes hold in three spatial dimensions, and also compute the emergent quantum critical phenomenon in the Yukawa-Lorentz symmetric hyperplane.

## 1.2 Organization

The rest of the paper is organized as follows. In the next section (Sec. 2), we introduce the lattice model for NH (pseudo)spin-3/2 birefringent Dirac fermions, and derive at their continuum description. In Sec. 3, we discuss the symmetry protection of such gapless NH quasiparticles and classify various insulating mass orders. Sec. 4 is devoted to the discussion on the emergent Yukawa-Lorentz symmetry from a leading order RG analysis within the framework of the $\varepsilon$ expansion. Summary of our findings, concluding remarks, and discussion on related issues are presented in Sec. 5. Additional symmetry protection of NH birefringent Dirac fermions is discussued in Appendix A. Technical details of our RG analysis are relegated to Appendix B, C, and D.

## 2 Lattice model

We begin the discussion by constructing a lattice realization of NH birefringent Dirac fermions, displaying real energy eigenvalue spectrum over an extended parameter regime. To arrive there, we first consider the generalized $\pi$-flux square lattice model fostering Hermitian birefringent Dirac fermions with two effective Fermi velocities. The corresponding unit-cell is composed of four sites, belonging to A, B, C, and D sublattices, with the intra-unit-cell nearest-neighbor hopping pattern shown in Fig. 1(a). Accordingly, we define a four-component spinor

$$\Psi^{\top}(\boldsymbol{q}) = [c_{A}(\boldsymbol{q})\, c_{B}(\boldsymbol{q})\, c_{C}(\boldsymbol{q})\, c_{D}(\boldsymbol{q})]\,, \tag{1}$$

where $c_j(\boldsymbol{q})$ is the fermion annihilation operator on the sites belonging to the sublattice $j$ with momentum $\boldsymbol{q}$. The associated Bloch Hamiltonian reads as [7,8]

$$H = \sum_{\boldsymbol{q}} \Psi^{\dagger}(\boldsymbol{q})\, H_{\text{lattice}}(\boldsymbol{q})\, \Psi(\boldsymbol{q})\,, \tag{2}$$

Table 1: Transformation of three mass orders $M_{\mathrm{I}} \equiv M$, $\boldsymbol{M}_{\mathrm{II}}$, and $M_{\mathrm{III}}$, belonging to the CI, CII, and CIII families, respectively, under the non-spatial time-reversal ($\mathcal{T}_+$), anti-unitary particle-hole ($\mathcal{T}_-$), and unitary particle-hole (PH) symmetries and the four-fold rotational symmetry about the $z$ direction ($R^z_{\pi/2}$), when the mass matrix $M$ appearing in the construction of the non-Hermitian birefringent Dirac operator is $M_{\mathrm{I}}$. Here $\mathcal{K}$ is the complex conjugation, $\checkmark$ ($\times$) corresponds to even (odd) under symmetry operations, and 0 (1) represents scalar (vector) under $R^z_{\pi/2}$. This symmetry analysis is shown in details in Sec. 3. Therefore, a combination of non-spatial ($\mathcal{T}_+$, $\mathcal{T}_-$ and PH) and spatial ($R^z_{\pi/2}$) symmetries forbids nucleation of any constant mass order without breaking at least one of the symmetries of the non-interacting non-Hermitian system.

| Mass | Symmetries | | | |
|---|---|---|---|---|
| | $\mathcal{T}_+ = \kappa M$ | $\mathcal{T}_- = \kappa$ | PH $= M$ | $R^z_{\pi/2}$ |
| $M_{\mathrm{I}} \equiv M$ | $\checkmark$ | $\times$ | $\times$ | 0 |
| $\boldsymbol{M}_{\mathrm{II}} = (M^1_{\mathrm{II}}, M^2_{\mathrm{II}})$ | $\checkmark$ | $\checkmark$ | $\checkmark$ | 1 |
| $M_{\mathrm{III}}$ | $\times$ | $\checkmark$ | $\times$ | 0 |

where

$$H_{\mathrm{lattice}}(\boldsymbol{q}) = \begin{bmatrix} 0 & -t_- \cos(q_y a) & 0 & t_- \cos(q_x a) \\ -t_- \cos(q_y a) & 0 & t_+ \cos(q_x a) & 0 \\ 0 & t_+ \cos(q_x a) & 0 & t_+ \cos(q_y a) \\ t_- \cos(q_x a) & 0 & t_+ \cos(q_y a) & 0 \end{bmatrix}, \tag{3}$$

$a$ is the lattice spacing, $t_\pm = t(1 \pm \beta)$, $t$ is the hopping amplitude, and $\beta$ is the dimensionless birefringent parameter with $|\beta| < 1$. This model shows linear touching of the filled valence and empty conduction bands at four points in the Brillouin zone $(\pm\pi/(2a), \pm\pi/(2a))$, when the system is maintained at the half-filling. Out of four such points only one is inequivalent, which we choose to be at $\mathbf{K} = (1,1)\pi/(2a)$. Around this point the system supports gapless birefringent Dirac fermions, captured by the Hamiltonian

$$H^{\mathrm{cont}}_{\mathrm{BF}}(\boldsymbol{k}) = v_{\mathrm{H}} \left[ -\Gamma_{11} k_x + \Gamma_{31} k_y \right] - \beta v_{\mathrm{H}} \left[ \Gamma_{22} k_x + \Gamma_{01} k_y \right], \tag{4}$$

where $v_{\mathrm{H}} = ta$ bears the dimension of the Fermi velocity, $\boldsymbol{k} = \boldsymbol{q} - \mathbf{K}$, and the Hermitian matrices $\Gamma_{\mu\nu} = \sigma_\mu \otimes \sigma_\nu$ for $\mu, \nu = 0, \cdots, 3$ define the basis for all four-dimensional matrices. Here, $\{\sigma_\mu\}$ is the set of Pauli matrices and $\otimes$ corresponds to a tensor product. The energy spectra of $H^{\mathrm{cont}}_{\mathrm{BF}}(\boldsymbol{k})$ are $\pm E_\pm(\boldsymbol{k})$, where $E_\pm(\boldsymbol{k}) = v_{\mathrm{H}}(1 \pm \beta)|\boldsymbol{k}|$. The positive (negative) energy branch represents the conduction (valence) band. For $\beta = 1$, there are two flat bands at zero energy, which we do not delve into here.

Notice that the operators proportional to $v_{\mathrm{H}}$ and $\beta v_{\mathrm{H}}$ in Eq. (4) separately constitute two Dirac Hamiltonian in two spatial dimensions, each containing two mutually anticommuting matrices multiplying the planar components of momentum $k_x$ and $k_y$. However, these two Dirac operators commute with each other. Thus, $H^{\mathrm{cont}}_{\mathrm{BF}}(\boldsymbol{k})$ cannot be cast in a block-diagonal form, in which each block is two-dimensional. In other words, there exists no unitary operator ($U_{\mathrm{block}}$), such that $U^\dagger_{\mathrm{block}} H^{\mathrm{cont}}_{\mathrm{BF}}(\boldsymbol{k}) U_{\mathrm{block}}$ assumes the form of a block matrix $H_+(\boldsymbol{k}) \oplus H_-(\boldsymbol{k})$, where $H_+(\boldsymbol{k})$ and $H_-(\boldsymbol{k})$ are expressed in terms of two-dimensional Pauli matrices. Therefore, the minimal irreducible representation of $H^{\mathrm{cont}}_{\mathrm{BF}}(\boldsymbol{k})$ is four-dimensional. Birefringent Dirac fermions this way honor the celebrated Nielsen-Ninomiya fermion doubling theorem [31]. In

conventional spin-1/2 Dirac systems (recovered by setting $\beta = 0$), the Hamiltonian is composed of two copies of irreducible two-dimensional Hamiltonian, also yielding a minimal reducible four-component representation, as shown in the original no-go theorem. Since the irreducible representation of birefringent Dirac fermions is four dimensional, they constitute to a spin-3/2 system (namely, $2s + 1 = 4 \Rightarrow s = 3/2$).

We recognize that this system supports a mass order, which in the announced four-component spinor basis [Eq. (1)] is represented by a diagonal Hermitian matrix

$$M = \text{diag.}(-1, 1, -1, 1), \tag{5}$$

that anti-commutes with $H_{\text{lattice}}(\mathbf{q})$ and $H_{\text{BF}}^{\text{cont}}(\mathbf{k})$. On the generalized $\pi$-flux square lattice, this operator captures a density imbalance at each site away from the half-filling that assumes a quadrupolar pattern within the unit-cell while maintaining the overall charge neutrality of the system, as shown in Fig. 1(b). Its uniform expectation value $\Delta$ (say) leads to an insulation of the system, as can be seen from the eigenvalues of the associated effective single-particle Hamiltonian $H_{\text{BF}}^{\text{cont}}(\mathbf{k}) + \Delta M$, given by $\pm\left[E_{\pm}^2(\mathbf{k}) + \Delta^2\right]$. However, nucleation of such a mass order requires spontaneous lifting of symmetry, which we will discuss in the next section. For now, in terms of $M$, we define an anti-Hermitian operator

$$H_{\text{lattice}}^{\text{AH}}(\mathbf{q}) = M H_{\text{lattice}}(\mathbf{q}) = \begin{bmatrix} 0 & t_- \cos(q_y a) & 0 & -t_- \cos(q_x a) \\ -t_- \cos(q_y a) & 0 & t_+ \cos(q_x a) & 0 \\ 0 & -t_+ \cos(q_x a) & 0 & -t_+ \cos(q_y a) \\ t_- \cos(q_x a) & 0 & t_+ \cos(q_y a) & 0 \end{bmatrix}, \tag{6}$$

by exploiting the fact that the product of two mutually anti-commuting Hermitian matrices is an anti-Hermitian matrix. Finally, by combining $H_{\text{lattice}}(\mathbf{q})$ and $H_{\text{lattice}}^{\text{AH}}(\mathbf{q})$, we promote the lattice model for NH birefringent Dirac fermions, explicitly given by

$$\begin{aligned} H_{\text{lattice}}^{\text{NH}}(\mathbf{q}) &= H_{\text{lattice}}(\mathbf{q}) + \alpha\, H_{\text{lattice}}^{\text{AH}}(\mathbf{q}) \\ &= \begin{bmatrix} 0 & -t_-^- \cos(q_y a) & 0 & t_-^- \cos(q_x a) \\ -t_-^+ \cos(q_y a) & 0 & t_+^- \cos(q_x a) & 0 \\ 0 & t_+^- \cos(q_x a) & 0 & t_+^- \cos(q_y a) \\ t_-^+ \cos(q_x a) & 0 & t_+^+ \cos(q_y a) & 0 \end{bmatrix}, \end{aligned} \tag{7}$$

where $t_+^{\pm} = t_+(1 \pm \alpha)$ and $t_-^{\pm} = t_-(1 \pm \alpha)$. The dimensionless parameter $\alpha$ measures the strength of non-Hermiticity, capturing the imbalance in the hopping amplitudes between the nearest-neighbor sites in the opposite directions, as shown in Fig. 1(c). In the continuum limit around the **K** point, the NH operator for birefringent Dirac fermions reads as

$$H_{\text{BF,NH}}^{\text{cont}}(\mathbf{k}) = H_{\text{BF}}^{\text{cont}}(\mathbf{k}) + \alpha M H_{\text{BF}}^{\text{cont}}(\mathbf{k}). \tag{8}$$

The eigenvalues of this operator are $\pm E_{\pm}^{\text{NH}}(\mathbf{k})$, where $E_{\pm}^{\text{NH}}(\mathbf{k}) = v_{\text{F}}(1 \pm \beta)|\mathbf{k}|$ with $v_{\text{F}} = \sqrt{v_{\text{H}} - v_{\text{NH}}^2}$ and $v_{\text{NH}} = \alpha v_{\text{H}}$, which are purely real as long as $|\alpha| < 1$. We restrict ourselves within this parameter regime.

Before closing this section, we point out that a $\pi$-flux cubic lattice model with only nearest-neighbor hopping amplitudes among spinless fermions gives birth to minimal eight-component massless Dirac fermions in three spatial dimensions [32]. A similar generalization of such a cubic-lattice model with an eight-site unit-cell accommodates three-dimensional birefringent spin-3/2 gapless Dirac fermions [9, 15]. Such a model allows a staggered density mass order, similar to the one shown in Eq. (5), however, assuming an octupolar pattern within the eight-site unit-cell. Hence, the above general principle of construction can immediately be applied to realize lattice regularized NH birefringent spin-3/2 Dirac fermions in three dimensions, which we do not show here explicitly.

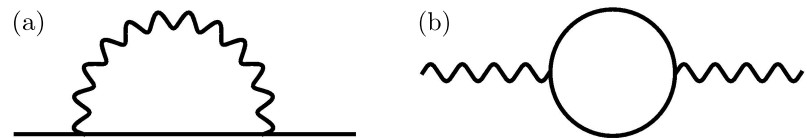

Figure 2: Leading-order (one-loop) self-energy diagrams for (a) non-Hermitian spin-3/2 birefringent Dirac fermions (solid lines) and (b) scalar bosonic order-parameter fields (wavy lines). The interaction vertex corresponds to the Yukawa interaction between these two degrees of freedom.

# 3 Symmetries and masses

The construction of the NH birefringent spin-3/2 Dirac operators [Eqs. (7) and (8)] requires introduction of a mass matrix $M$ [Eq. (5)] in its anti-Hermitian part. As any mass term breaks at least one of the symmetries of the free fermion system, it is natural to raise the question regarding the stability of gapless NH spin-3/2 Dirac fermions against the nucleation of any uniform mass order that causes an insulation in the system upon acquiring a finite vacuum expectation value. In this section, we show that a combination of non-spatial and rotational symmetries protects the nodal nature of such a collection of NH birefringent Dirac fermions. To this end, we consider a unitarily equivalent continuum model $U^{\dagger} H_{\mathrm{BF,NH}}^{\mathrm{cont}}(\boldsymbol{k}) U \to H_{\mathrm{BF,NH}}^{\mathrm{cont}}(\boldsymbol{k})$, where

$$U = \exp\left(i\frac{\pi}{4}\Gamma_{21}\right)\exp\left(i\frac{\pi}{4}\Gamma_{20}\right)\exp\left(i\frac{\pi}{4}\Gamma_{02}\right)\Gamma_{13}, \tag{9}$$

and after the unitary rotation

$$H_{\mathrm{BF}}^{\mathrm{cont}}(\boldsymbol{k}) = v_{\mathrm{H}}\left[\Gamma_{10}k_x + \Gamma_{30}k_y\right] + \beta v_{\mathrm{H}}\left[\Gamma_{01}k_x + \Gamma_{03}k_y\right]. \tag{10}$$

Then, $H_{\mathrm{BF,NH}}^{\mathrm{cont}}(\boldsymbol{k})$ assumes the from of Eq. (8) in terms of the unitarily rotated mass operator $M = \Gamma_{22}$. In this representation both $H_{\mathrm{BF}}^{\mathrm{cont}}(\boldsymbol{k})$ and $M$ are purely real, making our symmetry analysis simpler, which we discuss next.

In general, a NH operator $H_{\mathrm{NH}}$ (say) can be classified in terms of its transformations under (1) the time-reversal symmetry

$$\mathcal{T}_+ H_{\mathrm{NH}} \mathcal{T}_+^{-1} = H_{\mathrm{NH}}, \quad \text{and} \quad \mathcal{C}_+ H_{\mathrm{NH}}^{\dagger} \mathcal{C}_+^{-1} = H_{\mathrm{NH}}, \tag{11}$$

(2) the anti-unitary particle-hole symmetry

$$\mathcal{T}_- H_{\mathrm{NH}} \mathcal{T}_-^{-1} = -H_{\mathrm{NH}}, \quad \text{and} \quad \mathcal{C}_- H_{\mathrm{NH}}^{\dagger} \mathcal{C}_-^{-1} = -H_{\mathrm{NH}}, \tag{12}$$

(3) the unitary particle-hole (PH) symmetry symmetry

$$\mathrm{PH}\, H_{\mathrm{NH}}\, \mathrm{PH}^{-1} = -H_{\mathrm{NH}}, \tag{13}$$

and (4) the pseduo-Hermiticity (PSH) symmetry

$$\mathrm{PSH}\, H_{\mathrm{NH}}^{\dagger}\, \mathrm{PSH} = H_{\mathrm{NH}}, \tag{14}$$

detailed in Ref. [33]. When $H_{\mathrm{NH}} = H_{\mathrm{BF,NH}}^{\mathrm{cont}}(\boldsymbol{k})$, we find

$$\mathcal{T}_+ = M\mathcal{K}, \ \mathcal{T}_- = \mathcal{K}, \ \text{and} \ \mathrm{PH} = M, \tag{15}$$

where $\mathcal{K}$ is the complex conjugation, such that $\mathcal{T}_+^2 = +1$ (as it should be for spinless fermions), $\mathcal{T}_-^2 = +1$, and $\mathcal{K}\boldsymbol{k} \to -\boldsymbol{k}$. But, there are no representatives for $\mathcal{C}_+$, $\mathcal{C}_-$ and PSH. Thus, the NH operator $H_{\mathrm{BF,NH}}^{\mathrm{cont}}(\boldsymbol{k})$ does not possess $\mathcal{C}_+$, $\mathcal{C}_-$ and PSH symmetries. Besides these non-spatial

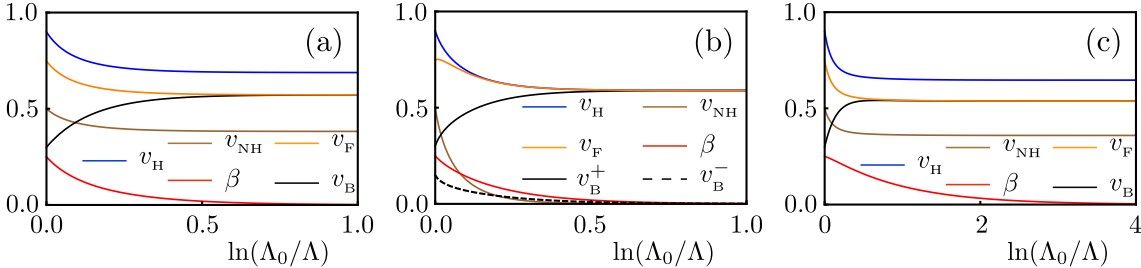

Figure 3: Renormalization group flows for various velocity (fermionic and bosonic) and birefringent ($\beta$) parameters close to the (a) CI, (b) CII, and (c) CIII mass generations, discussed in Sec. 4.1, Sec. 4.2, and Sec. 4.3, respectively, for $N_f = 1$, where $N_f$ is the number of four-component fermion species. In all these cases $\beta \to 0$ in the deep infrared or quantum critical regime, indicating that four-component spin-3/2 Dirac fermions decompose into two copies of two-component spin-1/2 Dirac fermions therein, where they possess a unique terminal Fermi velocity, which is equal to the terminal bosonic velocity, thereby always yielding an emergent Yukawa-Lorentz symmetry. In (a) and (c), we find $v_F = v_B$ with nontrivial $v_H$ and $v_{NH}$ (non-Hermitian Yukawa-Lorentz symmetry). By contrast, in (b) $v_H = v_B = v_B^+ = (v_{B_x} + v_{B_y})/2$ while $v_{NH} \to 0$ (conventional or Hermitian Yukawa-Lorentz symmetry), where $v_{B_x}$ ($v_{B_y}$) is the component of the bosonic velocity in the $x$ ($y$) direction. Near the CII mass generation the rotational symmetry gets broken at intermediate scales as $v_B^- = (v_{B_x} - v_{B_y})/2$ is finite therein, which however gets recovered in the quantum critical regime where $v_B^- \to 0$. Here, $\Lambda_0$ ($\Lambda$) is the bare (running) cutoff and $\ln(\Lambda_0/\Lambda) \equiv \ell$ with $\ell$ as the logarithm of the renormalization group scale. Throughout, we set $v_H^0 > v_{NH}^0$, where the quantities with the superscript '0' indicate their bare values, such that $v_F^0 = \sqrt{[v_H^0]^2 - [v_{NH}^0]^2}$ is always real, yielding real eigenvalues of the non-Hermitian birefringent Dirac operators. Here, we showcase emergent Yukawa-Lorentz symmetry when $v_F^0 > v_B^0$. We arrive at qualitatively similar conclusions for $v_F^0 < v_B^0$ as well (not shown explicitly).

symmetries, $H_{BF,NH}^{cont}(\boldsymbol{k})$ also remains invariant after a four-fold or $C_4$ rotation about the $z$ axis, generated by

$$R_{\pi/2}^z = \exp\left(i\frac{\pi}{4}\Gamma_{02}\right)\exp\left(i\frac{\pi}{4}\Gamma_{20}\right), \qquad (16)$$

under which $\boldsymbol{k} \to (k_y, -k_x)$. It should be noted that such a four-fold rotational symmetry is not an artifact of the continuum limit of the tight-binding model, rather results from a rotation of the whole lattice around a specific lattice site by an angle $\pi/2$ about the $z$ direction, which can be appreciated in the following way. For concreteness, we take a site belonging to the A sublattice as the center of rotation. Then after a rotation by $\pi/2$ about the A site, B $\to$ D, D $\to$ B, and X $\to$ X for X = A and C. Also recall that under $\pi/2$ rotation about the $z$ direction $(x, y) \to (y, -x)$ and $(A_x, A_y) \to (A_y, -A_x)$, where $\boldsymbol{A} = (A_x, A_y)$ is the magnetic vector potential, such that the magnetic field in the $z$ direction $B_z = (\nabla \times A)_z \to -(\nabla \times A)_z = -B_z$. In the Landau gauge chosen in our construction for the generalized $\pi$-flux square lattice model, as shown in Fig. 1(a), it corresponds to $t_- \to -t_-$. Under such relabeling of the sublattice indices and $t_- \to -t_-$, $H_{lattice}(\boldsymbol{q})$ from Eq. (3) remain invariant. Such a rotation also leaves the mass matrix ($M$) for the quadrupolar charge-density-wave mass order (discussed in details shortly), shown in Fig. 1(b), appearing in the construction of the NH birefringent Dirac operator invariant. Thus the tight-binding operator from Eq. (7) giving birth to NH birefringent Dirac fermions in the continuum limit, is invariant under the microscopic $C_4$ rotation.

In the absence of birefringence, conventional spin-1/2 Dirac systems accommodate four mass matrices [34], namely $\{\Gamma_{22}, \Gamma_{12}, \Gamma_{32}, \Gamma_{02}\}$, each of which fully anticommutes with $H_{\text{BF}}^{\text{cont}}(\boldsymbol{k})$ upon setting $\beta = 0$ therein [Eq. (10)]. When acquires a finite vacuum expectation value, each of them causes insulation in the system with an isotropically gapped quasiparticle spectrum. However, birefringence causes fragmentation among the mass orders and they can be grouped into three classes. To facilitate this classification, we decompose $H_{\text{BF}}^{\text{cont}}(\boldsymbol{k})$ as $H_{\text{BF}}^{\text{cont}}(\boldsymbol{k}) = v_{\text{H}} H_{01}(\boldsymbol{k}) + \beta v_{\text{H}} H_{02}(\boldsymbol{k})$, where $H_{01}(\boldsymbol{k}) = \Gamma_1 k_x + \Gamma_2 k_y$ and $H_{02}(\boldsymbol{k}) = \bar{\Gamma}_1 k_x + \bar{\Gamma}_2 k_y$ with $\Gamma_1 = \Gamma_{10}$, $\Gamma_2 = \Gamma_{30}$, $\bar{\Gamma}_1 = \Gamma_{01}$, and $\bar{\Gamma}_2 = \Gamma_{03}$. In this notation, which we also use in three appendices, four mass matrices fully anticommute with $H_{01}(\boldsymbol{k})$, and their classification in a birefringent Dirac system can be summarized in terms of the (anti)commutation relations with $H_{02}(\boldsymbol{k})$ [18].

Class I (CI) mass is represented by a single-component scalar order, given by $M_{\text{I}} = \Gamma_{22}$ that fully anticommutes with $H_{02}(\boldsymbol{k})$. Class II (CII) mass is a two-component vector order, given by $\boldsymbol{M}_{\text{II}} = (M_{\text{II}}^1, M_{\text{II}}^2) \equiv (\Gamma_{21}, \Gamma_{23})$. Each element of $\boldsymbol{M}_{\text{II}}$ commutes with one of the matrices ($\bar{\Gamma}_1$ and $\bar{\Gamma}_2$) appearing in the birefringent part of the Dirac Hamiltonian $H_{02}(\boldsymbol{k})$, while anticommuting with the other one. Class III (CIII) mass is also a scalar order, given by $M_{\text{III}} = \Gamma_{20}$ that fully commutes with $H_{02}(\boldsymbol{k})$. Transformations of these masses under the discrete non-spatial symmetries and four-fold rotational symmetry of $H_{\text{BF,NH}}^{\text{cont}}(\boldsymbol{k})$ are summarized in Table 1, when we choose $M \equiv M_{\text{I}}$ therein. Thus, none of the four mass matrices can gain any vacuum expectation value to trigger an insulation in the system without breaking at least one of the symmetries of $H_{\text{BF,NH}}^{\text{cont}}(\boldsymbol{k})$. Thus, gapless NH spin-3/2 Dirac excitations are symmetry protected. In Appendix A and Table 2, we show that no other fermion bilinear can acquire any finite vacuum expectation value unless at least one of the non-spatial and spatial symmetries of the non-interacting system is broken. While the lattice realization of CI mass is shown in Fig. 1(b), that for CII and CIII masses are shown in Fig. 2 of Ref. [18]. Notice that CI mass leads to quadrupolar arrangement of staggered fermionic density within a single plaquette while maintaining the overall charge neutrality of the system. On the other hand, the CII mass ordered phase fosters fermionic current between the nearest-neighbor sites either in the $x$ or in the $y$ direction, thus breaking the rotational symmetry. By contrast, CIII mass order allows fermionic current between the next-nearest-neighbor sites along the diagonal directions of the square lattice, such that the net flux through any plaquette is zero and the rotational symmetry is preserved.

# 4 Critical phenomena

Linear energy-momentum relation in $d$-dimensional Dirac systems gives rise to a vanishing density of states when $d > 1$, namely $\varrho(E) \sim |E|^{d-1}$ where $E$ is the energy. The same conclusion holds for NH birefringent Dirac systems. Consequently, any ordering sets in beyond a critical strength of short-range Hubbardlike interaction via a quantum phase transition. In this section, we investigate the emergent quantum critical phenomena near such phase transitions, with a special emphasis on the resulting Yukawa-Lorentz symmetry in its vicinity. Even though a Dirac system can in principle foster a plethora of ordered phases [35,36], here we focus only on the mass orders, discussed in the previous section. Mass orders cause insulation in the system, thereby yielding a maximal gain of the condensation energy at zero temperature (no competition with entropy).

We unfold the underlying quantum critical phenomena within the framework of an appropriate effective Gross-Neveu-Yukawa quantum field theory. The associated imaginary time ($\tau$) action is composed of three pieces and is given by $S = S_{\text{F}} + S_{\text{BF}} + S_{\text{B}}$. The fermionic action

takes the form

$$S_\text{F} = \int \Psi^\dagger(\tau, \boldsymbol{r})\Big[\partial_\tau + H^\text{cont}_\text{BF,NH}(\boldsymbol{k} \to -i\boldsymbol{\nabla})\Big]\Psi(\tau, \boldsymbol{r}), \tag{17}$$

where $\int \equiv \int d\tau d^d\boldsymbol{r}$, and $\Psi^\dagger(\tau, \boldsymbol{r})$ and $\Psi(\tau, \boldsymbol{r})$ are two independent Grassmann variables. The fermionic Green's function in terms of the Matsubara frequency ($\omega$) is

$$G_\text{F}(i\omega, \boldsymbol{k}) = \Big[i\omega + H^\text{cont}_\text{BF,NH}(\boldsymbol{k})\Big] \times \frac{\Big[\omega^2 + v_\text{F}^2(1+\beta)^2\boldsymbol{k}^2 - 2\beta v_\text{F}^2 H_{01}(\boldsymbol{k})H_{02}(\boldsymbol{k})\Big]}{(\omega^2 + v_\text{F}^2(1+\beta)^2\boldsymbol{k}^2)(\omega^2 + v_\text{F}^2(1-\beta)^2\boldsymbol{k}^2)}. \tag{18}$$

The Euclidean action for the self-interacting real bosonic order-parameter fields $\Phi_\alpha \equiv \Phi_\alpha(\tau, \boldsymbol{r})$ has the form

$$S_\text{B} = \sum_{\alpha=1}^{N_b} \int \left\{ \frac{1}{2}\Big[(\partial_\tau\Phi_\alpha)^2 + v_\text{B}^2(\partial_j\Phi_\alpha)^2 + m_\text{B}^2\Phi_\alpha^2\Big] + \frac{\lambda}{4!}(\Phi_\alpha^2)^2 \right\}, \tag{19}$$

where $N_b = 1(2)$ for CI and CIII (CII), $v_\text{B}$ is the velocity of the bosonic order-parameter field, $m_\text{B}^2$ is the tuning parameter for the transition, and $\lambda$ is the coupling constant for the $\Phi^4$ interaction. The Green's function for free bosonic fields in the critical hyperplane ($m_\text{B}^2 = 0$) is

$$G_{\text{B},\alpha\beta}(i\omega, \boldsymbol{k}) \equiv G_\text{B}(i\omega, \boldsymbol{k})\delta_{\alpha\beta} = \frac{1}{\omega^2 + v_\text{B}^2\boldsymbol{k}^2}\,\delta_{\alpha\beta}. \tag{20}$$

The Yukawa coupling between gapless birefringent excitations and bosonic order-parameter field takes the form

$$S_\text{BF} \equiv S_{\text{BF},j} = g\sum_{\alpha=1}^{N_b} \int \Phi_\alpha \Psi^\dagger(\tau, \mathbf{r})M_j^\alpha\Psi(\tau, \mathbf{r}), \tag{21}$$

as the latter is a composite object of two fermionic fields, where $j = \text{I}, \text{II}, \text{III}$, and $M^1_\text{I,III} \equiv M_\text{I,III}$. Here, $g$ is the Yukawa coupling. We extract the leading-order RG flow equations by computing the one-loop Feynman diagrams, shown in Fig. 2. The matrix algebra is performed in a fixed $d$. Subsequently, we integrate out the modes with Matsubara frequency $-\infty < \omega < \infty$. Finally, the momentum integral is perform in $d = 3 - \varepsilon$ around the upper critical three spatial dimensions, where the coupling constants $g$ and $\lambda$ are marginal, to capture the logarithmic divergences, appearing as $1/\varepsilon$ poles [37, 38]. These calculations are detailed in Appendix B, C, and D. Next, we discuss the RG flow equations for various velocity and birefringent parameters, and the emergent Yukawa-Lorentz symmetry near CI, CII, and CIII mass generations in the next three subsections, respectively. Notice that the forms of the fermionic and bosonic Green's functions result from their corresponding exchange statistics, which are same in both Hermitian and NH systems. Also the action is NH even in Hermitian systems. Thus, standard Feynman diagram approach can be pursued to unfold the behavior of interacting NH birefringent Dirac fermions in the deep infrared regime.

## 4.1 Near class I mass generation

Computation of the one-loop Feynman diagrams shown in Fig. 2 close to the CI mass generation, detailed in Appendix B, leads to the following RG flow equations also known as the

$\beta$-functions

$$\frac{dQ}{d\ell} = -2N_b g^2 Q \left[ \frac{2(v_F - v_B)(v_F + v_B)^2 + 4\beta^2 v_F^3}{3v_F v_B [(v_F + v_B)^2 - \beta^2 v_F^2]^2} \right], \tag{22}$$

$$\frac{d\beta}{d\ell} = -4N_b g^2 \beta \left[ \frac{(4v_F + v_B)v_B^2 + (4v_B + v_F(1-\beta^2))v_F^2}{3v_F v_B [(v_F + v_B)^2 - \beta^2 v_F^2]^2} \right], \tag{23}$$

$$\frac{dv_B}{d\ell} = -\frac{g^2 N_f}{v_F^3} \frac{v_B}{2} \left[ \frac{1 + 3\beta^2}{(1-\beta^2)^3} - \frac{v_F^2}{v_B^2} \frac{3 + 2\beta^2}{3(1-\beta^2)} \right], \tag{24}$$

for $Q = v_H$ and $v_{NH}$ after taking $g^2 k^{-\varepsilon}/(8\pi^2) \to g^2$, where $N_b = 1$ and $\ell$ is the logarithm of the RG scale. Here, $N_f$ is the number of four-component fermion species with $N_f = 1$ in our construction. These RG flow equations imply that at the Yukawa quantum critical point with $g^2 \sim \varepsilon$, the terminal velocities are such that $v_F = \sqrt{v_H^2 - v_{NH}^2} = v_B$, with all three velocities being nonzero, along with $\beta = 0$, independent of their initial values. We confirm this outcome by numerically solving the flow equations. See Fig. 3(a). Therefore, a new fixed point with an enlarged symmetry emerges, at which spin-3/2 Dirac fermions decompose into two copies of spin-1/2 Dirac fermions, while the system remains coupled to the environment and the single effective Fermi velocity of NH Dirac fermions ($v_F$) is equal to the bosonic order-parameter velocity ($v_B$), with the nonzero terminal values for $v_H$, $v_{NH}$, and $v_B$. We name it non-Hermitian Yukawa-Lorentz symmetry. As the mass matrix $M$ appearing in the anti-Hermitian part of the birefringent Dirac operator commutes with $M_I$, it is a member of the commuting class mass family, for which we arrive at a similar conclusion previously reported for spin-1/2 Dirac system [29], obtained by setting $\beta = 0$ from the outset.

## 4.2 Near class II mass generation

When a collection of NH birefringent Dirac fermions acquires a strong propensity toward the nucleation of CII mass order, contributions from the Feynman diagrams shown in Fig. 2 lead to the following RG flow equations

$$\frac{dv_H}{d\ell} = -2N_b g^2 v_H \left[ \frac{2(v_F - v_B)(v_F + v_B)^2 + 4\beta^2 v_F^3}{3v_F v_B [(v_F + v_B)^2 - \beta^2 v_F^2]^2} \right], \tag{25}$$

$$\frac{dv_{NH}}{d\ell} = -4N_b g^2 v_{NH} \left[ \frac{(2v_F + v_B)(v_F + v_B)^2 + 2\beta^2 v_F^3}{3v_F v_B [(v_F + v_B)^2 - \beta^2 v_F^2]^2} \right], \tag{26}$$

$$\frac{d\beta}{d\ell} = -2N_b g^2 \beta \left[ \frac{(v_F + v_B)^2(v_F + 2v_B) - v_F^3 \beta^2}{3v_F v_B [(v_F + v_B)^2 - \beta^2 v_F^2]^2} \right], \tag{27}$$

$$\frac{dv_B^\tau}{d\ell} = -g^2 \frac{N_f}{2v_F^3} \frac{v_B^\tau}{6} \left[ X(v_H, v_F, \beta) - \frac{\tau v_F^2}{10(v_B^+ + v_B^-)(1-\beta^2)} \left\{ \frac{Y(x,\beta)}{v_B^+ - v_B^-} - \tau \frac{\beta^2 Z(x,\beta)}{2v_B^\tau} \right\} \right], \tag{28}$$

for $\tau = \pm$ and $N_b = 2$, after taking $g^2 k^{-\varepsilon}/(8\pi^2) \to g^2$, where $v_B^\pm = (v_{B_x} \pm v_{B_y})/2$, $v_{B_x}$ ($v_{B_y}$) is the $x$ ($y$) component of the bosonic velocity in two dimensions, $x = (v_H^2 + v_{NH}^2)/v_F^2$, and

$$X(v_H, v_F, \beta) = 6 \frac{v_H^2}{v_F^2} \frac{1 + \frac{3}{2}\beta^4 - \frac{1}{2}\beta^6}{(1-\beta^2)^3},$$

$$Y(x,\beta) = 10 - 2\beta^2 + 3\beta^4 + x(50 - 18\beta^2) + 3\beta^4,$$

$$Z(x,\beta) = 8 - 12x - 2(1+x)\beta^2. \tag{29}$$

The details of the calculation are shown in Appendix C. Therefore, at an intermediate scale the bosonic velocity becomes anisotropic, as $v_{\mathrm{B}}^{-}$ is finite therein. However, as we approach the deep infrared or quantum critical regime $v_{\mathrm{B}}^{-} \to 0$, and $v_{\mathrm{B}}^{+} \to v_{\mathrm{B}} = v_{\mathrm{H}}$ with $v_{\mathrm{NH}} \to 0$ along with $\beta \to 0$. These outcomes are shown in Fig. 3(b) by numerically solving the above RG flow equations. Therefore, near the Gross-Neveu-Yukawa critical point ($g^2 \sim \varepsilon$), the system recovers full Hermiticity by decoupling itself from the environment and a conventional Yukawa-Lorentz symmetry emerges with $v_{\mathrm{NH}} = 0$ and $v_{\mathrm{F}} = v_{\mathrm{H}} = v_{\mathrm{B}}$, irrespective of their bare values. Notice that CII mass operator $M_{\mathrm{II}}$ anticommutes with $M$, appearing in the anti-Hermitian part of the NH birefringent Dirac operator. Hence, it belongs to the family of anticommuting class masses, for which we find analogous emergent conventional Yukawa-Lorentz symmetry in spin-1/2 Dirac systems as well [29].

### 4.3 Near class III mass generation

Computation of the Feynman diagrams shown in Fig. 2 close to the CIII mass generation, detailed in Appendix D, leads to the following RG flow equations

$$\frac{dQ}{d\ell} = -4 N_b g^2 Q \left[ \frac{(v_{\mathrm{F}} - v_{\mathrm{B}})(v_{\mathrm{F}} + v_{\mathrm{B}})^2 + 2\beta^2 v_{\mathrm{F}}^3}{3 v_{\mathrm{F}} v_{\mathrm{B}} [(v_{\mathrm{F}} + v_{\mathrm{B}})^2 - \beta^2 v_{\mathrm{F}}^2]^2} \right], \tag{30}$$

$$\frac{d\beta}{d\ell} = -2 N_b g^2 \beta \left[ \frac{2(v_{\mathrm{F}} + v_{\mathrm{B}}) v_{\mathrm{B}}^2}{3 v_{\mathrm{F}} v_{\mathrm{B}} [(v_{\mathrm{F}} + v_{\mathrm{B}})^2 - \beta^2 v_{\mathrm{F}}^2]^2} \right], \tag{31}$$

$$\frac{dv_{\mathrm{B}}}{d\ell} = -\frac{g^2 N_f}{v_{\mathrm{F}}^3} \frac{v_{\mathrm{B}}}{2} \left[ 1 - \frac{v_{\mathrm{F}}^2}{v_{\mathrm{B}}^2} \frac{3 - 6\beta^2 + \beta^4}{3(1 - \beta^2)} \right], \tag{32}$$

for $Q = v_{\mathrm{H}}$ and $v_{\mathrm{NH}}$ after taking $g^2 k^{-\varepsilon}/(8\pi^2) \to g^2$ with $N_b = 1$. As the CIII mass $M_{\mathrm{III}}$ commutes with $M$, it also belongs to the commuting class mass family. As a result, the outcomes are similar to the ones we previously discussed for CI mass in Sec. 4.1. Numerical solutions of the flow equations support this claim. See Fig. 3(c).

Before closing this section, we point out that near CI and CIII mass orderings, the fermionic self-energy diagram produces the following bilinear term

$$\beta \frac{2 N_b (v_{\mathrm{F}} + v_{\mathrm{B}}) g^2}{v_{\mathrm{B}} \left[ (v_{\mathrm{F}} + v_{\mathrm{B}})^2 - v_{\mathrm{F}}^2 \beta^2 \right]^2} \left( \sum_{j=1}^{2} \Psi^{\dagger} (i \Gamma_{j0} \Gamma_{0j}) \Psi \right) \frac{1}{\varepsilon}. \tag{33}$$

Similar term also gets generated in Hermitian systems for which we take $v_{\mathrm{F}} \to v_{\mathrm{H}}$ [18]. This term being proportional to $\varepsilon^{-1}$ acquires a divergent contribution and thus, in principle, should be included in the bare action. However, it is proportional to the birefringent parameter $\beta$, which vanishes under RG in the deep infrared regime, where we also recover the Lorentz symmetry. Hence, the generated term also disappears in this regime and does not affect any outcomes, we discussed so far. We also note that no such fermion bilinear term gets generated close to CII mass generation.

## 5 Summary and discussions

To summarize, here we formulate lattice regularized symmetry protected gapless spin-3/2 NH birefringent Dirac fermions, featuring a real eigenvalue spectrum that scales linearly with momentum with two effective Fermi velocities. Namely, we show that a hopping imbalance in the opposite directions between each pair of the nearest-neighbor sites on a two-dimensional

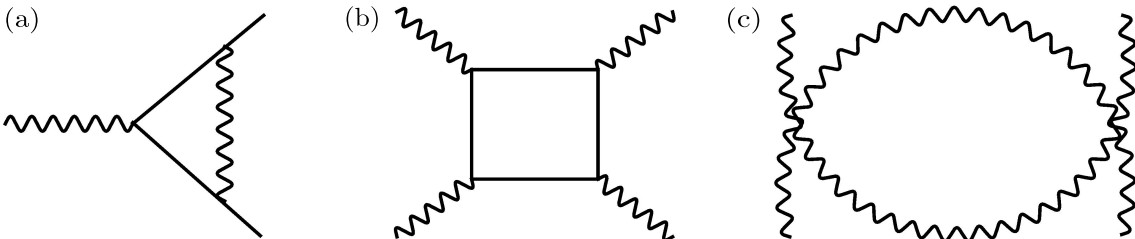

Figure 4: Feynman diagrams, yielding leading-order corrections for (a) the Yukawa interaction through the vertex correction, the $\Phi^4$ coupling due to (b) the Yukawa interaction, and (c) the bosonic interaction ($\lambda$) [37], leading to the RG flow equations in Eq. (34). Here solid (wavy) lines correspond to fermionic (scalar bosonic) field.

generalized $\pi$-flux square lattice with a four-site unit-cell fosters a collection such of unconventional quasiparticles. This construction is also directly applicable to a three-dimensional generalized $\pi$-flux cubic lattice model.

Such NH birefringent Dirac operators can be engineered at least on two-dimensional optical $\pi$-flux square lattices by combining the proposed scheme for its Hermitian counterpart [39, 40] and a recent suggestion to implement a left-right hopping imbalance on a one-dimensional chain [41]. Consider two copies of the generalized $\pi$-flux square optical lattice, occupied by neutral atoms living in the ground state and first excited state, which are coupled by running waves along four nearest-neighbor bond directions and the sites constituted by the excited state atoms undergo a rapid loss (the source of non-Hermiticity resulting from the coupling with the environment, yielding the loss). When the wavelength of the running wave is equal to the lattice spacing, the proposed two-dimensional NH birefringent Dirac operator can be realized on planar optical lattices. The requisite $\pi$ flux can be generated on optical lattices from artificial gauge fields [42, 43]. With the recent progress in realizing three-dimensional optical cubic lattices [44], this proposal can in principle be engineered there as well.

Strong Hubbardlike local interactions bring such systems close to quantum phase transitions, across which they become insulators with a gapped quasiparticle spectrum via the nucleation of mass orders. However, irrespective of the nature of the ordered state, the quantum critical state always features a Lorentz symmetry due to the Yukawa-type interactions between gapless fermionic and bosonic order-parameter fields. It manifests via a unique terminal velocity for both the participating degrees of freedom. This outcome has a profound consequence in the fermionic sector. With the disappearance of the birefringent parameter, spin-3/2 Dirac fermions decompose into two decoupled copies of Dirac fermions, transforming under the fundamental spin-1/2 representation. Recall that all the fermions in the Standard model transform under the fundamental spin-1/2 representation [45], which here we recover as an emergent phenomenon even for NH birefringent spin-3/2 Dirac fermions in the quantum critical regime. However, depending on the (anti)commuting Clifford algebra between the anti-Hermitian birefringent Dirac operator and the candidate mass operator, the Yukawa-Lorentz symmetry is accomplished while maintaining the non-Hermiticity (for commuting class masses) or by gaining full Hermiticity (for anticommuting class masses). One natural extension of our present study can be the quantum electrodynamics for NH birefringent Dirac fermions, in which scalar bosonic fields are replaced by helical spin-1 photons. Notice that this theory still remains unexplored even in Hermitian birefringent Dirac systems. Nonetheless, from a recent study on quantum electrodynamics of spin-1/2 NH Dirac fermions [30], we expect that such a theory will also manifest emergent spacetime Lorentz symmetry in which $v_{\mathrm{F}} \to c$ (speed of light) while keeping both $v_{\mathrm{H}}$ and $v_{\mathrm{NH}}$ finite, but $\beta \to 0$ in the deep infrared regime. However, in two and three spatial dimensions $c$ corresponds to the speed of light

in vacuum ($c_0$) and in medium with $c < c_0$, respectively, as dynamic screening by fermions, causing a reduction of the speed of light in a medium, is operative only in $d = 3$.

Within the Yukawa-Lorentz symmetric hyperplane, the RG flow equations for the dimensionless Yukawa coupling ($g$) and $\Phi^4$ interaction coupling ($\lambda$) read as

$$\frac{dg^2}{d\ell} = g^2\big[\varepsilon - (2N_f + 4 - N_b)g^2\big] \text{ and } \frac{d\lambda}{d\ell} = \varepsilon\lambda - 4N_f g^2\big[\lambda - 6g^2\big] - \frac{N_b + 8}{6}\lambda^2, \quad (34)$$

respectively. To arrive at these RG flow equations, we compute additional Feynman diagrams, shown in Fig. 4, besides two self-energy diagrams from Fig. 2 in the Lorentz symmetric hyperplane [37]. For CI and CIII mass orders, the dimensionless coupling constants are defined as $g^2 k^{-\varepsilon}/(8\pi^2 v_{\text{F}}) \to g^2$ and $\lambda k^{-\varepsilon}/(8\pi^2 v_{\text{B}}) \to \lambda$ with $v_{\text{F}} = v_{\text{B}}$, and $N_b = 1$. On the other hand, for CII mass order $g^2 k^{-\varepsilon}/(8\pi^2 v_{\text{H}}) \to g^2$ and $\lambda k^{-\varepsilon}/(8\pi^2 v_{\text{B}}) \to \lambda$ with $v_{\text{H}} = v_{\text{B}}$, and $N_b = 2$. The resulting Yukawa quantum critical point is located at

$$(g_*^2, \lambda_*) = \left(\frac{\varepsilon}{2N_f + 4 - N_b}, \frac{3\varepsilon}{(N_b + 8)(2N_f + 4 - N_b)}\right.$$
$$\left. \times \left[4 - 2N_f - N_b + \sqrt{(4 - 2N_f - N_b)^2 + 16N_f(N_b + 8)}\right]\right), \quad (35)$$

where both fermionic and bosonic two-point correlation or Green's functions become anomalous, governed by the respective anomalous dimensions, given by

$$\eta_\Psi = 2N_f g_*^2, \quad \text{and} \quad \eta_\Phi = N_b g_*^2/2. \quad (36)$$

As such, close to the Lorentz symmetric fixed point fermionic Green's function scales as $G_{\text{F}}^{-1} \sim (\omega^2 + v_{\text{F}}^2 \boldsymbol{k}^2)^{(1-\eta_\Psi)/2}$. Therefore, when the Yukawa critical point is approached from the ordered side of the transition, the residue of the NH fermionic quasiparticle pole vanishes as $Z_\Psi \sim (m_{\text{F}})^{\eta_\Psi/2}$, where $m_{\text{F}}$ is the fermionic mass, and its ratio with the bosonic mass ($m_{\text{B}}$) also takes a universal value $m_{\text{B}}^2/m_{\text{F}}^2 \sim \lambda_*/g_*$. The RG flow equation for the relevant coupling at the Yukawa critical point reads

$$\beta_{m_{\text{B}}^2} = m_{\text{B}}^2\left[2 - 2N_f g^2 - \frac{N_b + 2}{6}\right], \quad (37)$$

that determines the correlation length exponent

$$\nu = \frac{1}{2} + \frac{N_f}{2}g_*^2 + \frac{2 + N_b}{24}\lambda_*. \quad (38)$$

In two dimensions ($\varepsilon = 1$) $g_*^2, \lambda_* \sim \varepsilon$, yielding non-mean-field critical exponents. Consequently, the underlying quantum critical phenomenon is non-Gaussian in nature and the quantum critical fluid corresponds to a non-Fermi liquid without any sharp quasiparticle excitations (due to the non-trivial anomalous dimensions). By contrast, in three dimensions ($\varepsilon = 0$) all the critical exponents acquire their mean-field values, namely $\nu = 1/2$, $\eta_\Psi = \eta_\Phi = 0$, as then $g_*^2 = \lambda_* = 0$ [37, 38, 46, 47]. The resulting quantum phase transition is therefore Gaussian in nature and the Yukawa-Lorentz symmetric quantum critical fluid is a marginal Fermi liquid, in which residues of NH fermionic and bosonic quasiparticle poles decay logarithmically slowly. The universal critical exponents also govern the scaling of the interband (IB) component of the (real) frequency ($\omega$) dependent zero temperature longitudinal optical conductivity at the Yukawa critical point according to

$$\sigma_\star^{\text{IB}}(\omega) = \left(Z_\Psi^2\right)_{g^2 = g_*^2} \sigma_0^{\text{IB}}(\omega), \quad (39)$$

which follows from the gauge invariance of the current-current correlator [48]. Here, $\sigma_0^{\text{IB}}(\omega)$ is the interband component of the longitudinal optical conductivity at zero temperature in noninteracting systems. Respectively, in two and three spatial dimensions it is given by $e_0^2 \pi N_f/(4h)$ and $e_0^2 N_f \omega/(6h v_{\text{F}})$, where $e_0$ is the external electronic test charge [14]. As $\left(Z_\Psi^2\right)_{g^2=g_*^2} < 1$, the optical conductivity gets reduced at the Lorentz symmetric Yukawa fixed point in comparison to that in noninteracting systems in $d = 2$, while such a correction decreases logarithmically slowly with decreasing frequency in $d = 3$ as $g_*^2 = 0$ therein.

Our field theoretic predictions regarding the generic emergence of the Yukawa-Lorentz symmetry and the associated quantum critical phenomena can be tested from lattice-based quantum Monte Carlo simulations and exact diagonalization, for example. These approaches have already been pursued for correlated Hermitian birefringent Dirac systems in two and three spatial dimensions without encountering the infamous sign problem for the Hubbard interaction in the former method [13, 15]. Recently, the same numerical method has been successfully applied to NH Dirac systems with a similar hopping imbalance in the opposite directions between the nearest-neighbor sites on graphene's honeycomb lattice in the presence of the on-site Hubbard interaction [49]. The nature of the Yukawa-Lorentz symmetry can be pinned from the difference between the real-space two-point correlation functions, computed in the opposite directions for any pair of nearest-neighbor sites near the quantum phase transition. Namely, for the NH (Hermitian) Yukawa-Lorentz symmetry this quantity should be finite (zero). In addition, mean-field analysis [9,16] can be useful to scrutinize the impact of the NH parameter on the requisite strength of microscopic interactions for the nucleation of various ordered phases and the competition among them. Altogether, the current work should open new research directions, unfolding the effects of electron-electron interactions on the global phase diagram of NH birefringent Dirac fermions.

## Acknowledgments

We are thankful to Vladimir Juričić and Sanjib Kumar Das for critical reading of the manuscript.

**Funding information** This work was supported by NSF CAREER Grant No. DMR-2238679 of B.R. and Dr. Hyo Sang Lee Graduate Fellowship from Lehigh University (S.A.M.).

## A  Symmetry transformation of fermion bilinears

In Sec. 3 and Table 1, we have shown that the combination of discrete $\mathcal{T}_+$, $\mathcal{T}_-$, and PH symmetries and spatial four-fold $C_4$ rotational symmetry forbids nucleation of any uniform mass order in the spectrum of NH birefringent Dirac fermions unless at least one of these symmetries is spontaneously broken. Recall that three classes of mass orders (CI, CII, and CIII) are described by four four-dimensional Hermitian matrices. In this appendix, we perform analogous symmetry analysis to show that as such no fermion bilinear can acquire any vacuum expectation value unless one of the above mentioned symmetries is sacrificed. We do not consider fermion density operator $\Psi^\dagger \Gamma_{00} \Psi$, which can only receive a finite expectation value if the system is externally doped away from the half-filling. The remaining eleven four-component Hermitian matrices, appearing in the corresponding fermion bilinear, can be grouped as

$$S = \Gamma_{02}, \quad \boldsymbol{V_1} = (\Gamma_{10}, \Gamma_{30}), \quad \boldsymbol{V_2} = (\Gamma_{11}, \Gamma_{31}, \Gamma_{13}, \Gamma_{33}), \quad \boldsymbol{V_3} = (\Gamma_{12}, \Gamma_{32}), \quad \boldsymbol{V_4} = (\Gamma_{01}, \Gamma_{02}). \quad \text{(A.1)}$$

The symmetry transformations of these operators are shown in Table 2, confirming that the collection of gapless and rotationally symmetric NH Dirac fermions is symmetry protected.

Table 2: Transformation of matrices defined in Eq. (A.1) for which the corresponding fermion bilinears do not transform as mass orders under the non-spatial time-reversal ($\mathcal{T}_+$), anti-unitary particle-hole ($\mathcal{T}_-$), and unitary particle-hole (PH) symmetries and the four-fold rotational symmetry about the $z$ direction ($R^z_{\pi/2}$), when the mass matrix $M$ appearing in the construction of the non-Hermitian birefringent Dirac operator is $M_{\mathrm{I}}$. Here $\mathcal{K}$ is the complex conjugation, $\checkmark$ ($\times$) corresponds to even (odd) under symmetry operations, and 0 (1) represents scalar (vector) under $R^z_{\pi/2}$. Therefore, a combination of non-spatial ($\mathcal{T}_+$, $\mathcal{T}_-$ and PH) and spatial ($R^z_{\pi/2}$) symmetries forbids nucleation of any constant fermion bilinear order without breaking at least one of the symmetries of the non-interacting non-Hermitian system. See also Table 1.

| Bilinear matrix | Symmetries | | | |
|:---:|:---:|:---:|:---:|:---:|
| | $\mathcal{T}_+ = \kappa M$ | $\mathcal{T}_- = \kappa$ | PH $= M$ | $R^z_{\pi/2}$ |
| $S$ | $\times$ | $\checkmark$ | $\times$ | 0 |
| $V_1$ | $\times$ | $\times$ | $\checkmark$ | 1 |
| $V_2$ | $\checkmark$ | $\times$ | $\times$ | 1 |
| $V_3$ | $\checkmark$ | $\checkmark$ | $\checkmark$ | 1 |
| $V_4$ | $\times$ | $\times$ | $\checkmark$ | 1 |

## B  Details of RG calculations near CI mass generation

In this appendix, we show the details of RG calculations, leading to the flow equations for various velocity parameters and the birefringent parameter close to the CI mass generation. The contribution from the one-loop fermionic self-energy diagram near a class I mass generation quantum critical point can be written as

$$
\begin{aligned}
\Sigma(i\nu, \boldsymbol{k}) &= g^2 \int \frac{d^d \boldsymbol{p}}{(2\pi)^d} \int_{-\infty}^{\infty} \frac{d\omega}{2\pi} M_{\mathrm{I}} G_{\mathrm{F}}(i\omega, \boldsymbol{p}) M_{\mathrm{I}} G_{\mathrm{B}}(i\nu - i\omega, \boldsymbol{k} - \boldsymbol{p}) \\
&= g^2 \int \frac{d^d \boldsymbol{p}}{(2\pi)^d} \int_{-\infty}^{\infty} \frac{d\omega}{2\pi} (i\omega - H_{\mathrm{NH}}(\boldsymbol{p})) \\
&\quad \times \frac{(\omega^2 + v_{\mathrm{F}}^2(1+\beta^2)p^2 - 2\beta v_{\mathrm{F}}^2 H_{01}(\boldsymbol{p}) H_{02}(\boldsymbol{p}))}{(\omega^2 + v_{\mathrm{F}}^2(1+\beta)^2 p^2)(\omega^2 + v_{\mathrm{F}}^2(1-\beta)^2 p^2)[(\nu-\omega)^2 + v_{\mathrm{B}}^2(\boldsymbol{k}-\boldsymbol{p})^2]},
\end{aligned}
\tag{B.1}
$$

where $H_{01}(\boldsymbol{k}) = \Gamma_j k_j$ and $H_{02}(\boldsymbol{k}) = \bar{\Gamma}_j k_j$, and summation over repeated spatial index $j$ is assumed. To compute the RG flow equations of various parameters, appearing in $H_{\mathrm{BF,NH}}^{\mathrm{cont}}(\boldsymbol{k})$, we only need the divergent (div) pieces of $\Sigma(i\nu, \boldsymbol{k})$, denoted by $\Sigma^{\mathrm{div}}(i\nu, \boldsymbol{k})$, which we compute using the dimensional regularization around three spatial dimensions with $d = 3 - \varepsilon$ and the method of minimal subtraction. Then the divergent contributions are identified as $1/\varepsilon$ poles [37,38]. As to the leading-order there is no overlapping divergence $\Sigma^{\mathrm{div}}(i\nu, \boldsymbol{k}) \equiv \Sigma^{\mathrm{div}}(i\nu, 0) + \Sigma^{\mathrm{div}}(0, \boldsymbol{k})$. After some straightforward but lengthy algebra and per-

forming the integral over the Matsubara frequency ($\omega$), we find

$$
\begin{aligned}
\Sigma^{\text{div}}(0, \boldsymbol{k}) &= -\frac{g^2}{4\nu_{\text{F}}\nu_{\text{B}}} \int \frac{d^d p}{(2\pi)^d} \sum_{\tau=\pm} \frac{H_{01}^{\text{NH}}(\boldsymbol{p}) + \tau\beta H_{02}^{\text{NH}}(\boldsymbol{p})}{|\boldsymbol{p}||\boldsymbol{k}-\boldsymbol{p}|[\nu_{\text{F}}(1+\tau\beta)|\boldsymbol{p}| + \nu_{\text{B}}|\boldsymbol{k}-\boldsymbol{p}|]} \\
&= g^2 \frac{k^{-\varepsilon}}{4\pi^2\varepsilon} \bigg[ -\frac{(\nu_{\text{F}}+\nu_{\text{B}})^2(\nu_{\text{F}}+2\nu_{\text{B}}) - \nu_{\text{F}}^3\beta^2}{3\nu_{\text{F}}\nu_{\text{B}}[(\nu_{\text{F}}+\nu_{\text{B}})^2 - \beta^2\nu_{\text{F}}^2]^2} H_{01}^{\text{NH}}(\boldsymbol{k}) \\
&\quad + \frac{4\nu_{\text{F}}\nu_{\text{B}} + 3\nu_{\text{B}}^2 - \nu_{\text{F}}^2(\beta^2-1)}{3\nu_{\text{B}}[(\nu_{\text{F}}+\nu_{\text{B}})^2 - \beta^2\nu_{\text{F}}^2]^2} \beta H_{02}^{\text{NH}}(\boldsymbol{k}) \bigg] + \mathcal{O}(1),
\end{aligned}
\tag{B.2}
$$

where $H_{0j}^{\text{NH}}(\boldsymbol{k}) = \nu_{\text{H}} H_{0j}(\boldsymbol{k}) + \nu_{\text{NH}} M H_{0j}(\boldsymbol{k})$ for $j=1$ and 2. Following similar steps, we find the divergent piece of the fermionic self-energy diagram for zero external momentum to be

$$
\begin{aligned}
\Sigma^{\text{div}}(i\nu, 0) &= g^2 \int \frac{d^d p}{(2\pi)^d} \int_{-\infty}^{\infty} \frac{d\omega}{2\pi} (i\omega - H_{\text{NH}}(\boldsymbol{p})) \\
&\quad \times \frac{(\omega^2 + \nu_{\text{F}}^2(1+\beta^2)p^2 - 2\beta\nu_{\text{F}}^2 H_{01}(\boldsymbol{p})H_{02}(\boldsymbol{p}))}{(\omega^2 + \nu_{\text{F}}^2(1+\beta)^2 p^2)(\omega^2 + \nu_{\text{F}}^2(1-\beta)^2 p^2)((\nu-\omega)^2 + \nu_{\text{B}}^2 p^2)} \\
&= (i\nu) \frac{g^2 \nu^{-\varepsilon}}{4\pi^2\varepsilon} \frac{(\nu_{\text{F}}+\nu_{\text{B}})^2 + \nu_{\text{F}}^2\beta^2}{\nu_{\text{B}}[(\nu_{\text{F}}+\nu_{\text{B}})^2 - \beta^2\nu_{\text{F}}^2]^2} + \mathcal{O}(1).
\end{aligned}
\tag{B.3}
$$

We then obtain the following renormalization conditions

$$
Z_\psi + \frac{g^2 \nu^{-\varepsilon}}{4\pi^2\varepsilon} B(\nu_{\text{F}}, \nu_{\text{B}}, \beta) = 1, \qquad Z_\psi Z_Q + \frac{g^2 k^{-\varepsilon}}{4\pi^2\varepsilon} L(\nu_{\text{F}}, \nu_{\text{B}}, \beta) = 1,
$$

$$
\text{and } Z_\psi Z_Q Z_\beta - \frac{g^2 k^{-\varepsilon}}{4\pi^2\varepsilon} M(\nu_{\text{F}}, \nu_{\text{B}}, \beta) = 1,
\tag{B.4}
$$

where $Q = \nu_{\text{H}}$ or $\nu_{\text{NH}}$ and

$$
B = \frac{(\nu_{\text{F}}+\nu_{\text{B}})^2 + \nu_{\text{F}}^2\beta^2}{\nu_{\text{B}}[(\nu_{\text{F}}+\nu_{\text{B}})^2 - \beta^2\nu_{\text{F}}^2]^2}, \qquad L = \frac{(\nu_{\text{F}}+\nu_{\text{B}})^2(\nu_{\text{F}}+2\nu_{\text{B}}) - \nu_{\text{F}}^3\beta^2}{3\nu_{\text{F}}\nu_{\text{B}}[(\nu_{\text{F}}+\nu_{\text{B}})^2 - \beta^2\nu_{\text{F}}^2]^2},
$$

$$
\text{and } M = \frac{4\nu_{\text{F}}\nu_{\text{B}} + 3\nu_{\text{B}}^2 - \nu_{\text{F}}^2(\beta^2-1)}{3\nu_{\text{B}}[(\nu_{\text{F}}+\nu_{\text{B}})^2 - \beta^2\nu_{\text{F}}^2]^2},
\tag{B.5}
$$

with $X \equiv X(\nu_{\text{F}}, \nu_{\text{B}}, \beta)$ for $X = B, L$, and $M$. From these conditions, we arrive at the RG flow equations for $\nu_{\text{H}}$, $\nu_{\text{NH}}$ and $\beta$, shown in Eqs. (22) and (23).

Next we proceed to compute the bosonic self-energy correction, given by

$$
\Pi_{\text{B}}(\nu, \boldsymbol{k}) = -\frac{g^2}{2} \int \frac{d^d p}{(2\pi)^d} \int_{-\infty}^{\infty} \frac{d\omega}{2\pi} \text{Tr}[M_{\text{I}} G_{\text{F}}(\omega, \boldsymbol{p}) M_{\text{I}} G_{\text{F}}(\nu+\omega, \boldsymbol{k}+\boldsymbol{p})].
\tag{B.6}
$$

Once again, we are interested only in the divergent part of $\Pi_{\text{B}}(\nu, \boldsymbol{k})$, denoted by $\Pi_{\text{B}}^{\text{div}}(\nu, \boldsymbol{k})$, which we identify as $1/\varepsilon$ pole by performing the momentum integral in dimensions $d = 3-\varepsilon$. As the one-loop bosonic self-energy diagram is also devoid of any overlapping divergences, $\Pi_{\text{B}}^{\text{div}}(\nu, \boldsymbol{k}) = \Pi_{\text{B}}^{\text{div}}(\nu, 0) + \Pi_{\text{B}}^{\text{div}}(0, \boldsymbol{k})$. For zero external momentum, after completing the trace (Tr) algebra and integration over the Matsubara frequency ($\omega$) we find

$$
\begin{aligned}
\Pi_{\text{B}}^{\text{div}}(\nu, 0) &= \frac{g^2}{2\nu_{\text{F}}^d} (4N_f) \int \frac{d^d p}{(2\pi)^d} \frac{2p[4p^2(1-\beta^2) + \nu^2]}{[4p^2(1+\beta)^2 + \nu^2][4p^2(1-\beta)^2 + \nu^2]} \\
&= -\left( \frac{g^2 N_f \nu^{-\varepsilon}}{8\pi^2\varepsilon} \right) \nu^2 \frac{1+3\beta^2}{\nu_{\text{F}}^3(1-\beta^2)^3} + \mathcal{O}(1),
\end{aligned}
\tag{B.7}
$$

where $N_f$ is the number of four-component fermions. In order to extract $\Pi_{\mathrm{B}}^{\mathrm{div}}(0,\boldsymbol{k})$, we expand the corresponding integrand to the quadratic order in $\boldsymbol{k}$ and subsequently perform the integral over $\boldsymbol{p}$ in $d = 3 - \varepsilon$, yielding

$$\Pi_{\mathrm{B}}^{\mathrm{div}}(0,k) = -\left(\frac{g^2 k^{-\varepsilon} N_f}{8\pi^2 \varepsilon v_{\mathrm{F}}^3}\right)\frac{3+2\beta^2}{3(1-\beta^2)}v_{\mathrm{F}}^2 k^2 + \mathcal{O}(1). \tag{B.8}$$

We then obtain the following renormalization conditions

$$Z_\varphi + \frac{g^2 N_f \, v^{-\varepsilon}}{8\pi^2 v_{\mathrm{F}}^3 \varepsilon}\frac{1+3\beta^2}{(1-\beta^2)^3} = 1, \quad \text{and} \quad Z_\varphi Z_{v_{\mathrm{B}}^2} + \frac{g^2 N_f \, k^{-\varepsilon}}{8\pi^2 v_{\mathrm{F}}^3 \varepsilon}\frac{3+2\beta^2}{3(1-\beta^2)}\frac{v_{\mathrm{F}}^2}{v_{\mathrm{B}}^2} = 1, \tag{B.9}$$

from which we obtain the RG flow equation for $v_{\mathrm{B}}$, shown in Eq. (24).

## C  Details of RG calculations near CII mass generation

In this appendix, we show the details of RG calculations, leading to the flow equations for various velocity parameters and the birefringent parameter close to the CII mass generation. The fermionic self-energy correction due to class II mass ordering takes the form

$$\Sigma(i\nu,\boldsymbol{k}) = g^2 \sum_{j=1}^{2}\int\frac{d^d p}{(2\pi)^d}\int_{-\infty}^{\infty}\frac{d\omega}{2\pi}M_{\mathrm{II}}^{j}G_{\mathrm{F}}(i\omega,\boldsymbol{p})M_{\mathrm{II}}^{j}G_{\mathrm{B}}(i\nu - i\omega,\boldsymbol{k}-\boldsymbol{p}). \tag{C.1}$$

As discussed in the previous appendix, we only compute the divergent pieces of the self-energy term for zero external frequency and momentum separately. After some lengthy, but straightforward algebra the first quantity reads as

$$\begin{aligned}\Sigma^{\mathrm{div}}(0,\boldsymbol{k}) &= -\frac{g^2}{2v_{\mathrm{F}}v_{\mathrm{B}}}\int\frac{d^d p}{(2\pi)^d}\sum_{\tau=\pm}\frac{\tilde{H}_{01}^{\mathrm{NH}}(\boldsymbol{p})}{|\boldsymbol{p}||\boldsymbol{k}-\boldsymbol{p}|[v_{\mathrm{F}}(1+\tau\beta)|\boldsymbol{p}|+v_{\mathrm{B}}|\boldsymbol{k}-\boldsymbol{p}|]}\\ &= 2\left(g^2\frac{k^{-\varepsilon}}{4\pi^2\varepsilon}\right)\left(-\frac{(v_{\mathrm{F}}+v_{\mathrm{B}})^2(v_{\mathrm{F}}+2v_{\mathrm{B}})-v_{\mathrm{F}}^3\beta^2}{3v_{\mathrm{F}}v_{\mathrm{B}}[(v_{\mathrm{F}}+v_{\mathrm{B}})^2-\beta^2 v_{\mathrm{F}}^2]^2}\right)\tilde{H}_{01}^{\mathrm{NH}}(\boldsymbol{k}) + \mathcal{O}(1),\end{aligned} \tag{C.2}$$

where $\tilde{H}_{0j}^{\mathrm{NH}}(\boldsymbol{p}) = v_{\mathrm{H}}H_{0j}(\boldsymbol{k})-v_{\mathrm{NH}}MH_{0j}(\boldsymbol{k})$ for $j = 1$ and 2. The divergent piece of the fermionic self-energy diagram for zero external momentum reads as

$$\begin{aligned}\Sigma^{\mathrm{div}}(i\nu,0) &= 2g^2\int\frac{d^d p}{(2\pi)^d}\int_{-\infty}^{\infty}\frac{d\omega}{2\pi}\frac{i\omega(\omega^2+v_{\mathrm{F}}^2(1+\beta^2)p^2)}{(\omega^2+v_{\mathrm{F}}^2(1+\beta)^2 p^2)(\omega^2+v_{\mathrm{F}}^2(1-\beta)^2 p^2)((\nu-\omega)^2+v_{\mathrm{B}}^2 p^2)}\\ &= 2(i\nu)\frac{g^2 v^{-\varepsilon}}{4\pi^2\varepsilon}\frac{(v_{\mathrm{F}}+v_{\mathrm{B}})^2+v_{\mathrm{F}}^2\beta^2}{v_{\mathrm{B}}[(v_{\mathrm{F}}+v_{\mathrm{B}})^2-\beta^2 v_{\mathrm{F}}^2]^2}) + \mathcal{O}(1).\end{aligned} \tag{C.3}$$

Upon taking into account the divergent contributions from the fermionic self-energy diagram, we follow the procedure outlined in the previous appendix to arrive at the flow equations for $v_{\mathrm{H}}$, $v_{\mathrm{NH}}$ and $\beta$, reported in Eqs. (25)-(27).

The contribution from the one-loop bosonic self-energy diagram in this case reads as

$$\Pi_{B}^{j}(\nu,k) = -\frac{g^2}{2}\int\frac{d^d p}{(2\pi)^d}\int_{-\infty}^{\infty}\frac{d\omega}{2\pi}\mathrm{Tr}[M_{\mathrm{II}}^{j}G_{\mathrm{F}}(\omega,\boldsymbol{p})M_{\mathrm{II}}^{j}G_{\mathrm{F}}(\nu+\omega,\boldsymbol{k}+\boldsymbol{p})], \tag{C.4}$$

for $j = 1$ and 2, representing two bosonic modes associated with a two-component order-parameter field in $d = 2$. The divergent part of this diagram for zero external momentum is readily found following the steps outlined in the last appendix, yielding

$$\Pi_B^{j,\text{div}}(\nu, 0) = -\left(\frac{g^2 \nu^{-\varepsilon}}{8\pi^2 \varepsilon}\right) \frac{v_H^2}{v_F^2} \frac{1 + \frac{3}{2}\beta^4 - \frac{\beta^6}{2}}{v_F^3 (1 - \beta^2)^3} \nu^2 + \mathcal{O}(1), \tag{C.5}$$

which is independent of $j$. The computation of the bosonic self-energy diagram for zero external frequency is somewhat tedious and lengthy, leading to

$$\Pi_B^{j,\text{div}}(0, \boldsymbol{k}) = -\frac{g^2}{8\pi^2 v_F^3} \frac{N_f}{60} [L_x^j k_x^2 + L_y^j k_y^2] v_F^2 \frac{k^{-\varepsilon}}{\varepsilon} + \mathcal{O}(1). \tag{C.6}$$

It manifests explicit rotational symmetry breaking at any intermediate scale by bosonic order parameter fluctuations, which is natural to expect as the corresponding ordered state also breaks the four-fold rotational symmetry, see Table 1. Here,

$$L_x^1 = L_y^2 = \frac{10 - 18\beta^2 + 7\beta^4 + x(50 - 42\beta^2 + 7\beta^4)}{1 - \beta^2},$$

$$\text{and } L_y^1 = L_x^2 = \frac{10 - 2\beta^2 + 3\beta^4 + x(50 - 18\beta^2 + 3\beta^4)}{1 - \beta^2}, \tag{C.7}$$

with $x = (v_H^2 + v_{\text{NH}}^2)/v_F^2$. The explicit rotational symmetry breaking manifests via different RG flow equations for the $x$ and $y$ components of the bosonic velocity, respectively denoted by $v_{B_x}$ and $v_{B_y}$. They can be extracted from the renormalization conditions following the steps shown in the previous appendix. In order to demonstrate the emergent rotational symmetry near the associated Yukawa quantum critical point, we extract the RG flow equations for the following combinations of the bosonic velocities $v_B^+ = (v_{B_x} + v_{B_y})/2$ and $v_B^- = (v_{B_x} - v_{B_y})/2$, shown in Eq. (28).

# D  Details of RG calculations near CIII mass generation

In this appendix, we show the details of RG calculations, leading to the flow equations for various velocity parameters and the birefringent parameter close to the CIII mass generation. The fermionic self-energy correction near the class III mass nucleation reads as

$$\begin{aligned}
\Sigma(i\nu, k) &= g^2 \int \frac{d^d p}{(2\pi)^d} \int_{-\infty}^{\infty} \frac{d\omega}{2\pi} M_{\text{III}} G_F(i\omega, \boldsymbol{p}) M_{\text{III}} G_B(i\nu - i\omega, \boldsymbol{k} - \boldsymbol{p}) \\
&= g^2 \int \frac{d^d p}{(2\pi)^d} \int_{-\infty}^{\infty} \frac{d\omega}{2\pi} (i\omega - H_{01}^{\text{NH}}(\boldsymbol{p}) + \beta H_{02}^{\text{NH}}(\boldsymbol{p})) \\
&\quad \times \frac{(\omega^2 + v_F^2(1 + \beta^2)p^2 + 2\beta v_F^2 H_{01}(\boldsymbol{p}) H_{02}(\boldsymbol{p}))}{(\omega^2 + v_F^2(1+\beta)^2 p^2)(\omega^2 + v_F^2(1-\beta)^2 p^2)((\nu - \omega)^2 + v_B^2(\boldsymbol{k} - \boldsymbol{p})^2)}.
\end{aligned} \tag{D.1}$$

Following similar steps, outlined in the previous two appendices we find

$$\begin{aligned}
\Sigma^{\text{div}}(0, \boldsymbol{k}) = g^2 \frac{k^{-\varepsilon}}{4\pi^2 \varepsilon} \Bigg[ &-\frac{(v_F + v_B)^2 (v_F + 2v_B) - v_F^3 \beta^2}{3 v_F v_B [(v_F + v_B)^2 - \beta^2 v_F^2]^2} H_{01}^{\text{NH}}(\boldsymbol{k}) \\
&-\frac{4 v_F v_B + 3 v_B^2 - v_F^2(\beta^2 - 1)}{3 v_B [(v_F + v_B)^2 - \beta^2 v_F^2]^2} \beta H_{02}^{\text{NH}}(\boldsymbol{k}) \Bigg] + \mathcal{O}(1),
\end{aligned} \tag{D.2}$$

and

$$\Sigma^{\mathrm{div}}(i\nu, 0) = (i\nu)\frac{g^2 \nu^{-\varepsilon}}{4\pi^2 \varepsilon} \frac{(\nu_{\mathrm{F}} + \nu_{\mathrm{B}})^2 + \nu_{\mathrm{F}}^2 \beta^2}{\nu_{\mathrm{B}}[(\nu_{\mathrm{F}} + \nu_{\mathrm{B}})^2 - \beta^2 \nu_{\mathrm{F}}^2]^2} + \mathcal{O}(1). \tag{D.3}$$

From the divergent pieces of the fermionic self-energy diagram, we arrive at the RG flow equations for $\nu_{\mathrm{H}}$, $\nu_{\mathrm{NH}}$ and $\beta$, shown in Eqs. (30) and (31), following the steps outlined in Appendix B.

On the other hand, the one-loop bosonic self-energy diagram in this case takes the form

$$\Pi_B(\nu, k) = -\frac{g^2}{2} \int \frac{d^d p}{(2\pi)^d} \int_{-\infty}^{\infty} \frac{d\omega}{2\pi} \mathrm{Tr}[M_{\mathrm{III}} G_{\mathrm{F}}(\omega, \boldsymbol{p}) M_{\mathrm{III}} G_{\mathrm{F}}(\nu + \omega, \boldsymbol{k} + \boldsymbol{p})]. \tag{D.4}$$

The divergent pieces of the one-loop bosonic self-energy diagram are given by

$$\Pi_B^{\mathrm{div}}(\nu, 0) = -\left(\frac{g^2 \nu^{-\varepsilon} N_f}{8\pi^2 \varepsilon}\right) \nu^2 + \mathcal{O}(1),$$

$$\text{and} \quad \Pi_B^{\mathrm{div}}(0, \boldsymbol{k}) = -\left(\frac{g^2 k^{-\varepsilon} N_f}{8\pi^2 \varepsilon \nu_{\mathrm{F}}^3}\right) \frac{3 - 6\beta^2 + \beta^4}{3(1 - \beta^2)} \nu_{\mathrm{F}}^2 k^2 + \mathcal{O}(1). \tag{D.5}$$

Subsequently, we follow the steps outlined previously to obtain the renormalization conditions and the RG flow equation for $\nu_{\mathrm{B}}$, shown in Eq. (32).

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
