# Peer review of "Yukawa-Lorentz symmetry of interacting non-Hermitian birefringent Dirac fermions"

_SciPost Physics, doi:SciPost Phys. 18, 073 (2025)_

## Round 1 · Referee Report · Anonymous (Referee 1) · 2024-10-19

Strengths

I do not find any particular strength in the contents of this manuscript.

Weaknesses

  1. The paper argues that the introduction of pi-flux gives rise to nonreciprocal hoppings in the original lattice model, which results in a non-Hermitian (NH) Hamiltonian. However, the calculations later focus on a parameter regime where the eigenvalues pf the NH Hamiltonian take only real values. In that sense, the significance of non-Hermticity is quite obscure.

  2. For a NH Hamiltonian, I do not think there is a clear-cut machinery to set up the field theory and the consequent RG equations, which have all been developed for Hermitian systems. Hence, the entire set-up of the calculations suffer from fundamental issues.

  3. In recent times, it has become fashionable to extrapolate the systems studied in the past to some kind of NH description. This has become more of a fashionable terminology/jargon rather the real motive being extending our understanding of basic physics.

Report

The manuscript fails to fulfill any of the criteria that the authors have marked, namely, "Provide a novel and synergetic link between different research areas.",
"Open a new pathway in an existing or a new research direction, with clear potential for multi-pronged follow-up work",
"Detail a groundbreaking theoretical/experimental/computational discovery". It is the usual one-loop calculations leading to the RG equations of the couplings in the Hamiltonian, the technique which the last author has applied in many of his earlier works. The most problematic part is they have continued to use the same techniques for a system which they have claimed to represent NH physics, without any fundamental justification. Last but not the least, the results do not lead to any reasonable improvement of our knowledge of these kinds of systems.

Recommendation

Reject

  • validity: low
  • significance: poor
  • originality: low
  • clarity: ok
  • formatting: acceptable
  • grammar: good

Author:  Bitan Roy  on 2024-10-23  [id 4891]

(in reply to Report 1 on 2024-10-19)

We thank the Referee for the report. However, we are extremely disappointed by the quality of the report. We respectfully, but strongly disagree with all the criticisms drawn by the referee. Below we respond to each of them. In brief, we fail to find any scientific merit in this report.

Responses to "Weakness”:

1. We strongly disagree with the comment from the referee “However, the calculations later focus on a parameter regime where the eigenvalues of the NH Hamiltonian take only real values. In that sense, the significance of non-Hermiticity is quite obscure.”

The fact that a non-Hermitian (NH) operator supports purely real eigenvalues does not obscure the fact that the operator is NH. In fact, there is a surge of theoretical works geared to find guaranteed real eigenvalues for a NH operator, beginning from the seminal paper PRL 80, 5243 (1998). In this work, we proposed a general lattice model construction of NH operator that over an extended parameter regime shows real eigenvalue spectrum and we work in that parameter regime.

2. The comment from the referee “For a NH Hamiltonian, I do not think there is a clear-cut machinery to set up the field theory and the consequent RG equations, which have all been developed for Hermitian systems. Hence, the entire set-up of the calculations suffer from fundamental issues.” lacks any justification and is rather vague.

The referee could not point out a single step where our field theoretical approach suffers from any fundamental issue or flaw. We have presented all the technical details in the paper. We therefore invite the referee to spot a single step where our calculation encounters any fundamental issue.

3. We strongly disagree with the comment from the referee “In recent times, it has become fashionable to extrapolate the systems studied in the past to some kind of NH description. This has become more of a fashionable terminology/jargon rather the real motive being extending our understanding of basic physics.”

We pursued the project with a genuine fundamental goal: Establish the notion of emergent Lorentz symmetry for strongly interacting NH birefringent Dirac fermions. Our results extend the understanding and jurisdiction of fundamental physics in terms of emergent Lorentz symmetry to NH systems. We emphasize that the NH interacting model we analyze in this work can be simulated using quantum Monte Carlo technique, as has been done recently in PRL 132, 116503 (2024) for NH graphene-based model.

Responses to "Report”:

The referee criticized all three journal expectations that this manuscript fulfills according to the author without any justification. Below we present strong scientific arguments in favor of this claim by the authors.

1. “Provide a novel and synergetic link between different research areas” -- This is so because our work establishes the jurisdiction of emergent Lorentz symmetry in NH interacting birefringent Dirac systems, constituted by spin-3/2 fermions. Prior to the present work, any connection between Lorentz symmetry and NH higher-spin fermions was completely unknown.

2. “Open a new pathway in an existing or a new research direction, with clear potential for multi-pronged follow-up work” – This is so because our work establishes the quantum field theoretic techniques in NH systems, particularly in the new territory of NH higher-spin interacting fermions. The combination of non-Hermitian systems and quantum field theory constitutes a new and thriving future research direction that should attract attention from both condensed matter and high-energy physics communities. Furthermore, as the predicted results can be directly tested from quantum Monte Carlo simulations, for example, from our proposed concrete lattice models for NH birefringent fermions, as shown in PRL 132, 116503 (2024), it will generate “multi-pronged follow-up work”.

3. “Detail a groundbreaking theoretical/experimental/computational discovery” – This is so because the present work for the first time establishes the notion of Loretz symmetry and quantum criticality in interacting NH birefringent Dirac systems. While Lorentz symmetry plays the pivotal role in the unification of various fundamental forces, such a space-time symmetry prior to our work was known to be operative only in closed or Hermitian systems. We establish the footprints of such a venerable concept of theoretical physics in open or NH quantum systems, composed of spin-3/2 Dirac fermions.

Finally, it should be noted that application of an established formalism (quantum field theory, in our case) in a completely new system (NH birefringent Dirac systems, in our case) does not take away any novelty from a work. For example, the quantum Monte Carlo simulation is an old and well-developed technique. Still, it is being regularly used by researchers and the corresponding papers get published in high-impact peer-reviewed journals, such as Science, Physical Review Letters to name a few. The most relevant example of such publications in the present context is PRL 132, 116503 (2024). Hence, based on the existing scientific evidence, the comment from the referee “It is the usual one-loop calculations leading to the RG equations of the couplings in the Hamiltonian, the technique which the last author has applied in many of his earlier works.” does not diminish any scientific merit of the present work.

---

## Round 1 · Referee Report · Anonymous (Referee 2) · 2024-11-12

Strengths

See report.

Weaknesses

See report.

Report

The authors investigate symmetry-breaking quantum phase transitions in non-Hermitian 2D lattice models featuring birefringent Dirac-like fermionic excitations at low energies. The authors consider various candidate orders and study the effective (non-Hermitian) Gross-Neveu-Yukawa theories describing the quantum critical regime in each case. The main finding is that even though the microscopic model is neither Lorentz invariant nor Hermitian, at the critical point one finds emergent Lorentz invariance in the infrared, or emergent Hermiticity, or both.

To my knowledge the results of the paper are novel, and the finding that Hermiticity may or may not appear as an emergent “symmetry” at the quantum critical point is interesting.

However, before I can make an overall determination whether to recommend publication or not, I have a number of questions for the authors: 1. In the second paragraph after Eq. (4), the authors write that “these two Dirac operators commute with each other. Thus, H_{BF}^{cont}(k) cannot be cast in a block-diagonal form, in which each block is two-dimensional.” I do not understand this statement. 2. After Eq. (15), the authors write that “there are no representatives for C_+, C_-, and PSH”. If there are no representatives for those operators/matrices, how can they be bona fide symmetries? 3. The authors say that H_{BF,NH}^cont(k) is invariant under a C_4 rotation. I don’t see how the model can have C_4 symmetry if the hopping amplitude is different along the x and y directions (t_+ and t_-). Is it somehow a symmetry of the continuum limit, but not at the lattice scale? 4. Relatedly, from Eq. (B.6) it is not clear there is even C_4 symmetry in the continuum effective action, since the authors write that the bosonic self-energy “manifests explicit rotational symmetry breaking at any intermediate scale”. 5. The authors do not compute 1-loop diagrams for the vertex corrections (Yukawa and phi^4 vertices). Is it because they are only interested in the RG flow of velocities? Shouldn’t one in principle derive and solve the entire set of coupled RG equations, including the coupling constants? 6. Relatedly, have the authors checked that the effective action S contains all possible (relevant and marginal) terms allowed by symmetry, including those allowed by non-Hermiticity? Hermiticity (which becomes reflection positivity in Euclidean signature) highly constrains the possible terms, but once violated, I would expect there are many more possible terms. 7. The report by Referee 1 rightly (in my opinion) points out that it is not obvious that RG in non-Hermitian systems rests on the same rigorous footing as in Hermitian systems. I understand one can go through the mechanics of Feynman diagrams and read off renormalized coupling constants, but is there an overarching framework for non-Hermitian QFT that guarantees the methodology is sound? 8. To strengthen the motivation/overall impact of the work, it would be valuable if the authors could situate the current study within some larger framework, perhaps the general study/phenomenology of “non-Hermitian quantum criticality” (?). The authors briefly cite Refs. 29,30 by some of the same authors, but beyond specific models, is there something conceptually novel about quantum criticality in the non-Hermitian regime that doesn’t appear in the standard, Hermitian case? For example, are there novel universal observables one can compute (critical exponents, response functions) that are absent/meaningless in the Hermitian case? Does the present study of a specific model teach us something generic about quantum critical points in non-Hermitian systems? 9. A few specific comments regarding language/terminology: - At least in the abstract, it might be useful to use “pseudospin-3/2 Dirac fermions” instead of “spin-3/2 Dirac fermions” since Dirac fermions traditionally correspond to spin-1/2 representations of the Lorentz group. - In the first paragraph of the introduction, the sentence starting with “However, strongly interacting (via Hubbard-like local interactions, for example)…” is rather convoluted and should be reworded. - In the last sentence of the introduction, I do not understand what the authors mean by “establish the jurisdiction” of Lorentz symmetry. I suggest the authors use simpler and/or more scientific terms to convey their idea (“jurisdiction” is a legal term). - At the beginning of Sec. 1.1, the authors introduce the “masslike anti-Hermitian birefringent Dirac operator that also scales linearly with momentum”. If it scales linearly with momentum it is a kinetic term rather than a mass term. I suggest the authors remove “masslike” to avoid confusion. 10. Related to the last point, it would be useful if the authors could at least briefly discuss the three candidate orders in less abstract terms, i.e., what type of broken-symmetry phases do they correspond to in the microscopic model: is it a nematic phase, CDW order? Is the spectrum gapped or gapless? A schematic figure(s) may be a useful way to address this, but I would leave this up to the authors.

Requested changes

See report.

Recommendation

Ask for major revision

  • validity: good
  • significance: ok
  • originality: good
  • clarity: good
  • formatting: good
  • grammar: acceptable

Author:  Bitan Roy  on 2024-11-20  [id 4966]

(in reply to Report 2 on 2024-11-12)

We thank the referee for the report. In the first paragraph of the report, the referee compactly summarized the key outcomes of this work, which gives us confidence regarding the clarity of the presentation. A specific comment from the referee “To my knowledge the results of the paper are novel, and the finding that Hermiticity may or may not appear as an emergent “symmetry” at the quantum critical point is interesting.” is encouraging to the authors and endorses the novelty of this work. Below we respond to each comment/criticism and mention the intended changes in the revised resubmitted manuscript.

1.The flagged sentence means that the Hamiltonian from Eq. (4) cannot be expressed as direct sum of two copies of Dirac Hamiltonian which are written only in terms of the Pauli matrices. It can be verified that there exists no unitary transformation via which Eq. (4) can be cast into the form mentioned above. If this was the case, then we would have spin-1/2 Dirac fermions as is the case in graphene, instead of spin-3/2 birefringent Dirac fermions.

2.The flagged sentence implies that the NH birefringent Dirac operator does not possess C_+, C_-, and PHS. We first discuss all the non-spatial discrete symmetries that a generic NH operator can in principle possess, following Ref. 33. Then we check which symmetries are bona fide for the NH birefringent Dirac operator and which are not. In this case, it turns out that C_+, C_-, and PHS are not bona fide symmetries of this operator.

3.The C4 symmetry is a microscopic symmetry. But it is a bit challenging to explain it in words in a rebuttal without any equation. After Eq. (16) we substantially expand the discussion on the C4 symmetry to show that it is a symmetry at the microscopic level.

4.In Eq. (B.6) [now Eq. (C.6)] the C4 symmetry is broken by order-parameter fluctuations. This is perfectly allowed as any order-parameter describes a broken symmetry phase. Hence, it must break at least one of the symmetries of the free-fermion system. And class II (CII) mass breaks the C4 rotational symmetry, compatible with the symmetry analysis from Table 1.

5.To extract the RG flow equations of various velocities and the birefringent parameter, it is sufficient to compute the self-energy diagrams for fermions and bosons, shown in Fig. 2. However, to find the RG flow equations for the coupling constants, shown in Eq. (34), we need to compute other Feynman diagrams, such as the vertex diagram. Since those Feynman diagrams are well-known, shown in the book by Zinn-Justin (Ref. 37), we did not show them explicitly. We now show them as Fig. 4 and refer to it in the text.

6.If any term is allowed to be generated under RG, then it will get generated. In Table 1, we proved that no uniform mass term can be generated in the NH system unless some symmetry of non-interacting system is broken. In the new Table 2 and Appendix A, we show that no constant fermion bilinear term can be generated unless at least one symmetry is broken.

We realize that the RG procedure for CI and CIII masses generate fermion bilinear terms proportional to the birefringent parameter. As it is an irrelevant parameter, they do not alter any outcome. However, such terms do not get generated near CII mass generation.

7.The RG analysis is performed in terms of the fermionic and bosonic Green’s function. Their explicit forms depend only on the anticommutation and commutation relations of the fermionic and bosonic fields, respectively, which are identical in Hermitian and NH systems. Also action and fermionic Green’s function are NH even in a Hermitian system.

8.We presented sufficient broad discussion in 'Introduction’ and 'Summary and discussions’ sections. For example, we conjectured that emergent NH or Hermitian Lorentz symmetry should also be applicable in Dirac systems, where the bands are transforming under arbitrary spin-1/2 representation. We also proposed computation of a specific physical observable, namely the nearest-neighbor two-point correlation function in opposite direction to distinguish NH and Hermitian Lorentz symmetry. We now add a discussion on the signature of critical exponents on fermionic Green’s function and zero-temperature optical conductivity in 'Summary and discussion' and present some new comments on the broader implications of our results in 'Introduction' and 'Summary and discussions'. While such discussions are appealing, one should also contain them within a limit.

9.This set of comments concerns some stylistic changes. So, we respond to them in the "Changes in the revised manuscript" section below.

10.This comment also concerns stylistic changes. So, we respond to it in the "Changes in the revised manuscript" section below.

Changes in the revised manuscript (marked in BLUE color):

1.We clarify the statement flagged by the referee by adding discussion after this sentence.

2.After Eq. (15), we clarify that the NH birefringent Dirac operator does not possess the C_+, C_-, and PHS symmetries.

3.After Eq. (16), we expand the discussion to prove that C4 is a microscopic symmetry.

4.After Eq. (C.6) we clarify that CII mass breaks the C4 symmetry of the free-fermion system, which is compatible with Table 1, showing the symmetry analysis of all the mass terms.

5.We add a new figure (Fig. 4), showing the Feynman diagrams leading to the RG flow equations of the coupling constants, shown in Eq. (34), and refer to it in the text.

6.In a new Table 2 and Appendix A, we show that no fermion bilinear term with a uniform amplitude can be generated unless some symmetry of the non-interacting system is broken. We mention this result in the last paragraph of Sec. 3.

At the end of Sec. 4, we point out that during the RG procedure for CI and CIII mass nucleation, fermion bilinear terms get generated which are irrelevant and do not affect any outcome, reported in the manuscript. But no such term is generated for CII mass.

7.At the end of the paragraph after Eq. (21), we point out that the forms of the fermionic and bosonic Green’s functions depend on the anticommutation and commutation relations between the fermionic and bosonic fields, irrespective of whether the system is Hermitian or non-Hermitian. And thus, standard Feynman diagram techniques can be applied to unfold the infrared behavior of interacting NH systems.

8.In the 'Introduction' and 'Summary and discussions', we expand the discussion on the broad appeal of this work. In the 'Summary and discussions', we show the imprint of critical exponents on the fermionic Green’s function and zero-temperature optical conductivity.

9.In the abstract, we now introduce the term "pseudospin-3/2 Dirac fermions".

In the revised manuscript, we restructure the sentence "However, strongly interacting (via Hubbard-like local interactions, for example) …" flagged as convoluted by the referee.

Although the word "jurisdiction" is often used in legal context, in this case, "establish the jurisdiction of Lorentz symmetry" is appropriate as it means the applicability of the Lorentz symmetry. As authors of this manuscript, we must possess some degree of freedom to choose the wordings to present our own results.

The word "masslike" is necessary to appropriately characterize the structure of the anti-Hermitian term we add to the Hamiltonian to define the NH operator. Nevertheless, we properly define this term to remove any possibility of any confusion among the readers.

10.The physical meaning of one of the masses (class I or CI) is depicted in Fig. 1(b) and discussed in the text. We now add discussions on the physical natures of the CII and CIII masses, and their symmetry properties. However, we refrain from showing them as figures, as they are already available in Fig. 2 of Phys. Rev. Res. 2, 012047(R) (2020) [Ref. 18], which, we refer to in the discussion.

Attachment:

Response_Referee_SciPost_Birefringent.pdf

---

## Round 2 · Referee Report · Anonymous (Referee 2) · 2024-12-18

Report

The referees have significantly revised the manuscript and in doing so have addressed my concerns. I can recommend the paper for publication. (Please correct the typo "cpupling" in the caption of Fig. 4)

Recommendation

Publish (meets expectations and criteria for this Journal)

---

## Round 2 · Referee Report · Carlo Beenakker (Referee 3) · 2025-1-6

Strengths

  • remarkable finding of an emergent relativistic symmetry in an open quantum system
  • timely contribution to the active field of non-Hermitian quantum mechanics

Weaknesses

  • no immediate physical realisation available

Report

I was asked to report on this manuscript after two referees gave conflicting recommendations. The critical referee objected that the Hamiltonian, while being non-Hermitian, still had a real spectrum, which would obscure the impact of the non-Hermiticity. In the literature this class of Hamiltonians is actually of particular interest, since it allows to distinguish effects from the non-orthogonality of the eigenstates from more mundane effects of complex eigenvalues. I do agree with the critical referee that the application of the model studied here to an experimentally accessible system is uncertain, but I would not find this an objection that should stand in the way of publication: The paper studies a highly non-trivial problem of strongly interacting fermions in a system coupled to an environment, with implications for our fundamental physical understanding of the emergence of relativistic symmetries. I would consider a study of this type suitable for publication in SciPost Physics.

Recommendation

Publish (meets expectations and criteria for this Journal)

---

## Editorial Decision

published